

# The observed diurnal cycle of nocturnal low-level stratus clouds over southern West Africa: a case study

Karmen Babić[1], Bianca Adler[1], Norbert Kalthoff[1], Hendrik Andersen[1], Cheikh Dione[2],
Fabienne Lohou[2], Marie Lothon[2], and Xabier Pedruzo-Bagazgoitia[3]

[1]Institute of Meteorology and Climate Research, Karlsruhe Institute of Technology (KIT), Germany
[2]Laboratoire d'Aérologie, Université de Toulouse, CNRS, UPS, France
[3]Meterology and Air Quality Group, Wageningen University and Research, Wageningen, the Netherlands

*Correspondence to:* Karmen Babić (karmen.babic@kit.edu)

**Abstract.** This study presents the first detailed observational analysis of the complete diurnal cycle of stratiform low-level clouds (LLC) and involved atmospheric processes over southern West Africa. The data used here were collected during the comprehensive DACCIWA (Dynamics-Aerosol-Chemistry-Cloud-Interactions in West Africa) ground-based campaign, which aimed at monitoring LLC characteristics and capturing the wide range of atmospheric conditions related to the West African

monsoon flow. In this study, in-situ and remote sensing measurements from the supersite near Savè (Benin) collected during a typical day, which is characterized by the onset of a nocturnal low-level jet (NLLJ) and the formation of LLC, are analyzed. The associated dynamic and thermodynamic conditions allow the identification of five different phases of the LLC diurnal cycle: the Stable, Jet, Stratus I, Stratus II and Convective phase. The analysis of relative humidity tendency shows that cooling is a dominant process for LLC formation, which leads to a continuous increase of relative humidity at a maximum rate of 6

% per hour, until finally saturation is reached and LLC form with a cloud-base height near the height of NLLJ maximum. Results of heat budget analysis illustrate that horizontal cold air advection, related to the maritime inflow, which brings the cool maritime air mass and a prominent NLLJ wind profile, has the dominant role on the observed strong cooling of $-1.2$ K per hour during the Jet phase. The contribution from horizontal cold advection is quantified to be up to 72 %, while radiative cooling and sensible heat flux divergence contribute with 16 and 12 %, respectively, to the observed heat budget below the

NLLJ maximum. After the LLC form (Stratus phase I and II), turbulent mixing is an important factor leading to the cooling below the cloud base, while strong radiative cooling at the cloud top helps to maintain thick stratus.

## 1   Introduction

During the summer monsoon season in southern West Africa (SWA), stratiform low-level clouds (LLC) frequently form during

the night with typical cloud-base height (CBH) of several hundred meters above ground and cover extensive areas (Knippertz et al., 2011; Schrage and Fink, 2012; Schuster et al., 2013; van der Linden et al., 2015). Using multiyear (2006-2011) surface synoptic observations and various satellite products, van der Linden et al. (2015) presented a first climatology of the LLC during the wet monsoon season (July-September) in SWA. They found that shortly after sunset LLC frequently form along the



Guinea Coast and spread farther inland in the course of the night, while LLC are most frequent upstream of elevated terrain. The LLC reach a maximum northward extend between 09:00 and 10:00 UTC, with the maximum aerial coverage of approximately 800 000 km$^2$. Similar results were obtained by Schrage and Fink (2012) and Schuster et al. (2013), who analyzed the data collected during the African Monsoon Multidisciplinary Analysis (AMMA) 2006 special observing periods (May-October).

Due to their persistence until early afternoon hours, LLC significantly influence the radiation budget at the surface (Schuster et al., 2013; Hill et al., 2018), and thus affect the diurnal cycle of the atmospheric boundary layer (ABL) and regional climate (Knippertz et al., 2011).

So far, only few studies focused on the analysis of mechanisms and factors controlling the onset and maintenance of LLC. Schrage and Fink (2012) used remote sensing observations at Nangatchori in central Benin, Schuster et al. (2013) performed

regional simulations for the whole 2006 monsoon season using the Weather Research and Forecasting (WRF) model, while Adler et al. (2017) performed high-resolution numerical simulations with the Consortium for Small-Scale Modeling (COSMO) model for a case study and an area around the city of Savè (Benin). Schrage and Fink (2012) found the formation of LLC to be related to the onset of the nocturnal low-level jet (NLLJ) and the same conclusion is obtained from the model simulations. Schrage and Fink (2012) proposed the strong wind shear underneath the NLLJ, which leads to the destabilization of the near-

surface layer and increased turbulent upward mixing of cold air and moisture, to be the major process for the cloud formation at Nangatchori. On the other hand, modeling results of Schuster et al. (2013) suggest that turbulent processes related to the NLLJ are more dominant close to the coast, while orographically forced lifting on the windward side of mountains is more important farther inland. The importance of horizontal cold air advection with the south-westerly monsoon flow for the formation of LLC was found by both modeling studies (Schuster et al., 2013; Adler et al., 2017). Finally, after LLC form, radiative cooling at

the cloud top, as well as the vertical mixing below the cloud, help to maintain the stratus deck. High-resolution simulations suggest that additional processes could be important for LLC formation, such as vertical cold air advection, which is related to orographically induced lifting as well as to gravity waves, and enhanced convergence and upward motion upstream of existing clouds (Adler et al., 2017).

While LLC and involved phenomena are the integral part of the West African Monsoon system, climate models struggle to

realistically represent them. Knippertz et al. (2011) conducted a comprehensive analysis of global climate models and found positive biases in NLLJ speed, negative biases in LLC cover, and consequently large overestimation of solar radiation (of up to 90 W m$^{-2}$). Hannak et al. (2017) extended the analysis of Knippertz et al. (2011) to the latest global climate models data sets. While similar biases are found as in Knippertz et al. (2011), the authors have identified differences in subgrid cloud schemes as one of the possible reasons why models struggle to realistically represent LLC.

Up to now, spatial and temporal investigations of LLC in this region have been performed based mainly on satellite images, synoptic observations and few modeling studies mentioned above, while high-quality observational data sets were rare. Due to these limitations, processes which control the formation and dissolution of LLC are still not fully understood. Moreover, understanding of these processes has important practical implications, such as improving operational forecast and predictions of the West African monsoon in weather, seasonal and decadal climate simulations (Knippertz et al., 2011; Hannak et al.,

2017). In order to fulfill this gap, a comprehensive field campaign was conducted within the framework of the Dynamics-





aerosol-chemistry-cloud-interactions over West Africa (DACCIWA) project (Knippertz et al., 2015) in June and July 2016. The campaign included ground-based measurements at three supersites in Ghana, Benin and Nigeria (Kalthoff et al., 2018), radiosonde and aircraft measurements (Flamant et al., 2018). Knippertz et al. (2017) presented the large-scale setting, synoptic and mesoscale weather systems, which affected the region during the DACCIWA campaign, identifying different phases of

similar meteorological conditions, while Kalthoff et al. (2018) gave an overview of the diurnal cycle of the ABL conditions as well as of the conditions related to nocturnal LLC at three ground-based supersites.

This study is conducted in concert with the analyses presented by Adler et al. (2018) and Dione et al. (2018). In this study we focus on the description of the diurnal cycle of LLC and identification of physical processes and factors that control the formation, maintenance and dissolution of LLC during one typical night with undisturbed monsoon conditions and persistent

LLC (e.g. Flamant et al., 2018; Kalthoff et al., 2018). For this we use measurements performed at the Savè supersite during the intensive observation period (IOP) 8 (7–8 July 2016). Specifically, we aim at identifying the main factors leading to the relative humidity (RH) change and LLC formation and assessing the heat budget terms for different phases during the life-cycle of the LLC. Although we show only one case study, the dominant processes are considered to be representative for a major part of the DACCIWA campaign, especially for the *Post-onset phase* of monsoon (22 June–20 July 2016, Knippertz et al.,

2017). This is supported by the analysis of Adler et al. (2018), who perform in a consistent manner an analysis for 11 IOPs and we find a good agreement between these two studies. Additionally, Adler et al. (2018) investigated LLC characteristics (vertical extent, coverage, onset time and horizontal distribution and evolution), as well as the intra-night variability of boundary layer conditions and processes relevant for LLC formation. Dione et al. (2018) present a detailed statistical analysis on the characteristics of the LLC and the dynamics in the lower atmosphere for a 41 day period.

This paper is organized as follows: in Section 2 a brief description of the study site, data and methods used is given. In Section 3, the evolution of LLC is described, followed by the presentation of atmospheric dynamic and thermodynamic conditions in Section 4. In this section we also analyze the RH tendency and the heat budget. The discussion of results is presented in Section 5, while main findings are summarized in Section 6.

## 2   Data and methods

### 2.1   Meteorological measurements at Savè supersite

In this study we analyze the data collected during the DACCIWA ground-based measurement campaign, which took place between 14 June and 31 July 2016 at the Savè (Benin) supersite (Fig. 1). The comprehensive and unique data set of the ground-based campaign consists of remote sensing and in situ data (Bessardon et al., 2018), which enable the investigation of cloud characteristics, dynamic and thermodynamic conditions at high temporal and vertical resolutions (Kalthoff et al., 2018). During

the campaign, 15 IOPs were conducted during which, in addition to normal radiosondes launched at standard synoptic times, frequent radiosondes were launched at regular intervals in between the normal radiosondes. The Savè supersite ($8.00^o$ N, $2.43^o$ E, 166 m above sea level (a.s.l.)) is located approximately 185 km inland from the coast in a moderately hilly region which is favorable for LLC formation (van der Linden et al., 2015; Adler et al., 2017; Kalthoff et al., 2018). A comprehensive mea-





surement set-up at this supersite allows the detailed investigation of atmospheric conditions and identification of atmospheric processes relevant to the observed life-cycle of LLC.

Meteorological data used here comprise near-surface meteorological parameters, 30-min averaged turbulence fluxes and turbulence variables, which were calculated using the TK3.11 software (Mauder et al., 2013), and radiation fluxes (Kohler and

et. al., 2016).

The dynamic and thermodynamic conditions in the ABL were measured with radiosondes (normal and frequent) and different continuously running active and passive remote sensing instruments. The radiosondes were launched in regular intervals of 1.5 h, starting at 17:00 UTC prior to the IOP day (7 July) until 11:00 UTC on the IOP day. For Benin, local standard time equals UTC plus 1 hour. Additionally, the radiosounding measurements at the coastal station Accra (Ghana, Fig. 1) were performed

as part of the DACCIWA radisonde campaign (Flamant et al., 2018). High-resolution information of flow conditions (wind speed and direction) is obtained from a sodar (for lower part of the ABL) and an ultra-high-frequency (UHF) wind profiler (above 200 m a.g.l.) measurements. Additionally, Doppler lidar azimuth scans at $15^o$ elevation angle (plan-position indicator, PPI) provided the information on the three-dimensional radial velocity field by applying the velocity-azimuth display (VAD; Browning and Wexler, 1968) technique. In addition to the retrieval of the mean horizontal wind field with the VAD method,

PPI scans can be used to quantify turbulence (e.g. Bonin et al., 2017). We use PPI scans performed at $15^o$ elevation angle to estimate the variance of radial velocity for each range ring and project them to the vertical axis. The procedure is as follows: first we dismiss all range rings affected by clouds and remove outliers and echos by hard targets. Then we estimate the mean wind speed and direction using a simplified version of the VAD method (Bonin et al., 2017). The variance for each range ring is calculated by averaging the deviation from the fitted curve and corrected for uncorrelated noise. We obtained the information

on liquid water path (LWP) and integrated water vapor (IWV) from the microwave radiometer (Wieser et al., 2016) using the retrieval algorithm provided by the University of Cologne (Lönert and Crewell, 2003; Lönert et al., 2009). This algorithm was trained on a set of more than 12 000 radiosonde profiles measured between 1980 and 2014 in Abidjan (Ivory Coast). These quantities are needed for the radiative transfer model, which we used to obtain vertical profiles of radiative fluxes, and which is described in the Subsection 2.3.

## 2.2   Measurements of LLC characteristics

The cloud characteristics are documented by ceilometer (CBH), cloud radar measurements (cloud top height, CTH) and infrared (IR) cloud camera (cloud cover). The CBH is determined from the attenuated backscatter coefficient profiles based on a threshold method (manufacturer Lufft, personal communication, 2016), while the CTH is derived from the measurements of radar reflectivity of hydrometeors using a threshold of $-35$ dBz, i.e. reflectivities larger than $-35$ dBz are considered as clouds

(Bauer-Pfundstein and Goersdorf, 2007). In addition to this, information of sky conditions is obtained with the IR cloud camera. This camera takes images of the sky, which are coded in three colors: red, green and blue. The color of the image depends on the emissivity of the sky and, consequently, on the temperature. Thus, a red image indicates a relatively warm temperature, while blue indicates colder temperatures. Based on this, it is possible to distinguish cloud free periods, periods with continuous cloud deck, as well as periods with intermittent (stratus fractus) clouds.



The Spinning Enhanced Visible Infra-Red Imager (SEVIRI) data provided the spatiotemporal characteristics of LLC in the larger area. The spatiotemporal distribution and evolution of low clouds in the study area is observed with the SEVIRI (Schmetz et al., 2002) sensor, mounted on the geostationary Meteosat Second Generation satellite system. Information on LLC coverage is inferred using three channels: the visible at 0.6 $\mu$m (VIS), middle-infrared at 3.9 $\mu$m (MIR) and thermal-infrared

at 10.8 $\mu$m (TIR) at their native resolution of 3x3 km$^2$ (at nadir) and a repeat rate of 15 minutes. During daytime (06:00–18:00 UTC) the reflectance in the VIS channel is used to illustrate cloud coverage. Higher-level clouds are masked out by applying a TIR brightness temperature threshold at 283 K. Based on observed temperature profiles, this approximates a cloud-top altitude of 2.7 km. At night, LLC are illustrated using the brightness temperature difference of the TIR and the MIR channels. The underlying concept is based on the assumption that the average droplet size within the LLC is smaller than in higher-altitude

clouds. As the emissivity difference of the TIR and the MIR is dependent on cloud droplet size (Hunt, 1973), the brightness temperature difference between TIR and MIR has been used frequently to detect low clouds from various satellite platforms (e.g. Eyre and Allam, 1984; Cermak and Bendix, 2007). Additionally, the same TIR threshold is applied as during daytime to mask out higher-level clouds. It should be noted that edges of mid-level clouds might be missed by the TIR filter. The satellite imagery shown in this study is meant to be a purely qualitative representation of LLC occurrence and distribution during the

IOP.

## 2.3    SBDART radiative transfer model

Vertical profiles of radiative fluxes are computed with Santa Barbara DISORT Atmospheric Radiative Transfer (SBDART) model. SBDART is a software tool which computes plane-parallel radiative transfer in clear and cloudy conditions within the Earth's atmosphere and at the surface and the description of the program is given by Ricchiazzi et al. (1998). Thanks to the rich

data set obtained within the field campaign, many of the input parameters can be specified, allowing for a realistic modeling of radiative fluxes. For example, based on the radiosonde measurements, vertical profiles of pressure, temperature and water vapor are used in the model, while microwave radiometer provides IWV values. The standard tropical ozone density profile is used as input (linearly interpolated to model levels), since this information is not available from measurements. In total, 65 vertical input levels are specified, with 50 m resolution in the lowest 2.5 km and 1 km resolution between 3 and 16 km.

The spectral range of radiative flux calculations is selected to correspond to the measurement range of the near-surface solar radiation instrument, namely, between 0.34 and 2.2 $\mu$m in the shortwave range and 4.5–42 $\mu$m in the longwave range with spectral resolution of 0.01 and 0.1 $\mu$m, respectively. The solar illumination angles are computed from the specified day of year, time and geographic coordinates. A spectrally uniform albedo equal to 0.2 (Kalthoff et al., 2018) is set for the surface reflectance properties. The boundary layer aerosol type is set to typical rural, while the vertical optical depth of the boundary

layer aerosol is defined as a mean daily aerosol optical depth measured on July 7 (Level 1.5, http://aeronet.gsfc.nasa.gov/) and is equal to 0.36. In the case of cloud presence, cloud properties, such as cloud layer range and the optical thickness of the cloud layer, are specified. The cloud optical thickness is determined from the LWP measurements and the cloud droplet effective radius. We use the default value of cloud droplet effective radius of 8 $\mu$m as this value is within the range of aircraft measurements in the area (Deetz et al., 2018b). The phase function model used in cloud layers is mie scattering. All other input



## 3 Characteristics of the diurnal cycle of LLC

The SEVIRI-based information about the spatial distribution and temporal evolution of LLC is shown in Fig. 2. In the early evening, some patchy LLC are present in the investigation area and are confined to higher terrain (Fig. 1) of the Atakora Mountains range (Togo) and upstream of the Oshogbo Hills (Nigeria) (Fig. 2a, b). After 22:00 UTC, the first LLC formed southwest, i.e. upstream, of Savè and then extended to the downstream side by 00:00 UTC (Fig. 2c, d). At the same time, the area in the neighboring Nigeria covered with LLC was extending westwards until the two areas merge into one large area

around 01:00 UTC (Fig. 2d, e). At about 02:00 UTC, LLC have already extended and cover a substantial part of the domain (Fig. 2f), which continues to grow in the course of the night (Fig. 2g), so that at 05:00 UTC LLC cover the large part of the investigation area (Fig. 2h). After sunrise at around 06:00 UTC (Fig. 2i), LLC start slowly to dissipate (Fig. 2j, k), however at 11:00 UTC (Fig. 2l) their presence in the domain is still substantial.

The ceilometer backscatter measurements at Savè (Fig. 3a) show some low- and mid-level clouds (up to 3 km, not shown)

present between 18:00 and 22:00 UTC, followed by a cloud-free period. The LLC formed around midnight, with CBH at around 300 m a.g.l., and the same was observed by SEVIRI (Fig. 2d). These LLC are maintained during the rest of the night and even after sunrise, with CBH at approximately 250 m a.g.l. After around 08:00 UTC, the CBH rises approximately linearly with time. At first the CTH is observed roughly at 500 m a.g.l. indicating on average 250 m deep cloud layer, with a period between 01:00 and 03:00 UTC without a clear cloud radar signal, therefore making it difficult to determine the CTH for the

whole period (Fig. 3b). After 03:00 UTC, LLC are persistent until 08:00 UTC, with the CTH roughly constant at 650 m a.g.l. forming a 400 m deep cloud layer. After around 08:00 UTC, the CTH rises linearly as well. The microwave radiometer measurements of the LWP reveal varying cloud characteristics during the night (Fig. 3c). Apparently, in the first couple of hours LLC contain less liquid water, most likely due to the combination of a shallow cloud layer and lower optical thickness, and according to the sky conditions obtained with IR cloud camera, they are intermittent as well (Fig. 3d). After about 03:00

UTC, the LWP increases considerably probably because the cloud deepens, while a continuous cloud cover is observed. A minimum of IWV is observed just prior to the LLC onset and in the course of the night only a slight increase is observed. The observed differences in LLC characteristics suggest varying atmospheric conditions, therefore, we inspect dynamic and thermodynamic conditions at Savè in the next section.





## 4 Atmospheric conditions relevant for the diurnal cycle of LLC

### 4.1 Low-level jet and thermodynamic conditions

We start the investigation of atmospheric conditions during this IOP by inspecting the horizontal wind field (Fig. 4). The large-scale conditions on this IOP are characterized by about 1000-m-deep monsoon layer with southwesterly winds and the

African easterly jet above (Flamant et al., 2018). Note that on this particular IOP the observed monsoon depth is lower than for the whole DACCIWA investigation period, which has a median depth of 2 km (Dione et al., 2018). The minimum wind speed is found within the layer between about 1000 and 1500 m a.g.l., which corresponds to the transition layer between the southwesterly monsoon flow and easterlies above. In the African easterly jet layer, winds reach maximum of 15 m s$^{-1}$ at about 3500 m a.g.l. (see Fig. 1 in Dione et al., 2018). During the afternoon and early evening, a moderate northwesterly-to-

southwesterly flow of 3 m s$^{-1}$ prevails in the lowest 1500 m (Fig. 4a). The onset of NLLJ is observed at 20:30 UTC, with an abrupt increase of wind speed up to a maximum of 8 m s$^{-1}$, at a height of 275 m a.g.l. At the time of the NLLJ onset wind direction changes from westerly to southerly and south-southwesterly. The height of the NLLJ maximum corresponds to the height at which LLC form roughly 3.5 h later (Fig. 3b). Once the clouds have formed, the NLLJ maximum shifts upwards to the height of around 450 m a.g.l. reaching the maximum speed of 10 m s$^{-1}$ (Fig. 4a). A weakening of the NLLJ is seen

after 04:00 UTC and the axis is lifted to around 600–700 m a.g.l. With respect to wind speed close to the surface, we observe a similar behavior as in Lothon et al. (2008), with the wind speed only slightly increasing above 1.5 m s$^{-1}$ after the NLLJ onset (Fig. 4b). The wind direction becomes less variable after the arrival of south-southwesterly NLLJ and southwesterly flow persists in the course of the night and the following morning.

The potential temperature isolines show that after the sunset at 18:00 UTC, a strong cooling of the layer close to the ground

occurs, while coincident with the NLLJ onset this layer becomes deeper and reaches approximately 750 m depth (Fig. 4a). The period between the onset of NLLJ and the formation of LLC is characterized with the strongest decrease of temperature. After the continuous LLC deck has formed around 02:30 UTC, the temperature is roughly constant below the cloud base as well as within the cloud layer. The increase of the temperature coincides with the increase of the CBH after 08:00 UTC due to evolving convective boundary layer (CBL).

So far we have seen that due to the observed varying atmospheric conditions, the investigated period can be divided into different phases. The first phase identified is the period between the sunset (18:00 UTC) and the onset of the NLLJ (20:30 UTC), when the increasing static stability causes the decoupling of the mixed layer from the stable surface layer and this phase is denoted as *Stable phase*. This phase is followed by the *Jet phase*, a time period between the onset time of NLLJ and the formation of LLC (00:00 UTC), which marks the beginning of *Stratus phase*. The period of roughly 2.5 hours after

LLC formation is identified as the *Stratus phase I*, since inhomogeneous cloud cover and unstationary atmospheric conditions are observed. This is followed by a *Stratus phase II*, which corresponds to the period between 02:30 and 06:30 UTC, with a persistent LLC deck and quasi-stationary atmospheric conditions. The final *Convective phase* is associated with growing CBL and is characterized with increased surface heating and lifting of the cloud base. Note that these five phases do not occur only on this particular IOP. Adler et al. (2018) found that the same phases can be distinguished for at least 10 other IOPs.



The wind shear underneath the NLLJ causes mechanical production of turbulence, which is considered to be an important process leading to the LLC formation (Zhu et al., 2001; Schrage and Fink, 2012; Schuster et al., 2013). Figure 5a shows the absolute values of gradient Richardson number ($Ri$), which we calculate from radiosonde profile measurements for the 50-m averaged bins according to the following expression: $Ri = \frac{g}{\theta} \frac{\partial\theta/\partial z}{(\partial U/\partial z)^2}$, where $g$ is the acceleration due to gravity, $\theta$ is the potential temperature and $U$ is the horizontal wind speed. Generally, turbulence is stronger as the Richardson number is smaller, while $Ri = 0.21 - 0.25$ is consider to be a critical Richardson number below which the flow is fully turbulent. When $Ri$ is above 1, the flow is considered to be laminar (e.g. Stull, 1988). Although two Doppler lidars were deployed at Savè, we could not obtain reliable measurements of vertical velocity fluctuations ($\sigma_w$) from the vertical stare mode observations (Adler et al., 2018). However, for an assessment of turbulence in the nocturnal cloud-free ABL, radial velocity measurements during PPI scans performed with the scanning Doppler lidar at Savè can be used too. The standard deviation of the radial velocity ($\sigma_{rv}$) measured by Doppler lidar is shown in Fig. 5b.

The importance of the NLLJ for LLC formation was first reported by Schrage and Fink (2012) for the SWA region, while Zhu et al. (2001) found similar importance for the nocturnal stratus in the Great Plains (USA). The signature of the NLLJ in the near-surface measurements is expected to be mostly seen in the TKE and not necessarily in the mean wind speed, as suggested by Lothon et al. (2008) in their AMMA study.

Figure 5a, b shows that before the sunset, a CBL is still present at 17:00 UTC, while during the Stable phase turbulence decays due to the lack of mechanically generated mixing since the wind speed is rather low. Additionally, due to the longwave radiative cooling of the surface after sunset resulting in a negative sensible heat flux, the stable ABL develops and the negative buoyancy suppresses vertical mixing in the ABL, which is evident from low values of the TKE and high stability parameter ($\zeta$, Fig. 5c). Stability parameter is defined as the ratio of a height $z$ and the Obukhov length $L = -u_*^3/(k\frac{g}{\theta}\overline{w'\theta'})$, where $u_*$ is the friction velocity, $k = 0.4$ is von Kármán constant and $\overline{w'\theta'}$ is the kinematic heat flux. This parameter is traditionally used as a measure of stability in the surface layer. Its magnitude is not directly related to static stability, but positive sign indicates statically stable conditions and negative implies unstable (e.g. Stull, 1988). Simultaneously with the NLLJ onset, turbulence increases in the upper and lower shear zone of the jet. In the Jet phase, below the NLLJ maximum and at the surface static stability decreases enabling stronger turbulent mixing and increase of TKE (Fig. 5a, c), while simultaneously the sensible heat flux decreases from $-10$ W m$^{-2}$ to its maximum value of $-20$ W m$^{-2}$ during the night (not shown). In the first couple of hours after the LLC formed, mostly intermittent turbulence ($0.25 < Ri < 1$) is present within the LLC in the shear zones of the upper part of the jet (Fig. 5a), while increased turbulent mixing is evident below the CBH (Fig. 5a, b), which results in a highly turbulent ABL. We notice that there is a quite good agreement in the information about the turbulence intensity obtained from different measurement systems. The profiles of radial velocity variance show higher values in the lower and upper shear zones of the NLLJ maximum, as well as below the CBH.

Figure 6a, b shows temporal evolution of RH in combination with potential temperature and specific humidity. In the Stable phase the increase of RH is confined to the lowest 100 m, with the simultaneous decrease of temperature and increase of specific humidity. At the surface, temperature decreases by 4 $^o$C, while a small increase in specific humidity occurs ($\sim$ 1 g kg$^{-1}$), which finally leads to an increase of RH from 70 to 85 % (Fig. 6c). We observe the drop of temperature by 3 $^o$C in the





period between 20:00 and 00:00 UTC, while simultaneously RH increases by 10 % in the layer below 700 m a.g.l. We note that simultaneously with the RH increase after 20:00 UTC, an increase in ceilometer backsatter is observed as well (Fig. 3). This is most likely related to the aerosol hygroscopic growth, i.e. the size and composition of particles change due to their water vapor uptake (Deetz et al., 2018a; Haslett et al., 2018). The beginning of the Jet phase is associated with a slight increase of specific

humidity, which afterwards does not change considerably until 23:00 UTC, when it even decreases. After saturation has been reached, LLC form. The unsteady conditions during the subsequent roughly two to three hours (Stratus phase I) are reflected in the RH measurements, with the sonde released at 02:00 UTC not reaching saturation. At the same time, the decrease of temperature is accompanied with the decrease of moisture, i. e. specific humidity decreased at a rate of 0.5 g kg$^{-1}$ h$^{-1}$ below 600 m a.g.l. This suggests that the air mass behind the NLLJ is drier than the environment at Savè. After 03:00 UTC, the

conditions below the cloud base are quasi-stationary. The cooling of the near-surface layer weakens after the cloud decked forms, which leads to near-neutrally stratified surface layer ($\zeta \approx 0$, Fig. 5c) and contributes to further vertical mixing (Fig. 5c). After 08:00 UTC, RH starts to decrease with a simultaneous increase of temperature and specific humidity.

Based on radiosonde measurements, CBH and CTH can be determined by applying different criteria on RH measurements, such as the criteria described in Kalthoff et al. (2018). Their criteria detect cloud layers when RH is larger than 99 %. Com-

parison of RH profiles with CBH and CTH shown in Fig. 6a clearly shows that three radiosonde profiles would indicate a deeper cloud layer than detected by the cloud radar. While on average there is a good agreement between the CBH estimates from ceilometer and radiosondes, RH measurements can suggest a too high CTH due to the condensation on the sensor even after the sonde has risen above the cloud top. This issue highlights the advantage of the DACCIWA ground campaign and the multitude of instruments deployed allowing for the multiple estimates of certain parameters and their cross validation.

In the following sections we present the analysis of processes relevant for the evolution of LLC and asses their relevance during different phases.

## 4.2 Relative humidity tendency

In order to quantify whether the specific humidity ($q$) or temperature ($T$) change has a stronger influence on the RH tendency and consequently on LLC formation, we determine their respective contributions using consecutive radiosonde measurements.

The tendency of RH is calculated from the time derivative of $RH = e/e_s$, where $e$ is the water vapor pressure and $e_s$ is the saturation water vapor pressure. In the next step, we incorporate the Clausius-Clapeyron relation $\frac{\partial e_s}{\partial T} = \frac{L_v e_s}{R_v T^2}$ and the definition of water vapor pressure, $e = \frac{q}{0.378q + 0.622} p$, where $T$ is the air temperature in Kelvin, $L_v$ is the latent heat of vaporization $(2.5 \times 10^6$ J kg$^{-1})$ and $R_v$ is the universal gas constant. Finally, the contribution of absolute values and tendencies of $q$ and $T$ to RH tendency is calculated according to:

$$\underbrace{\frac{\partial RH}{\partial t}}_{(I)} = \underbrace{\frac{p}{e_s} \frac{0.622}{(0.378q + 0.622)^2} \frac{\partial q}{\partial t}}_{(II)} - \underbrace{\frac{p}{e_s} \frac{qL_v}{(0.378q + 0.622)R_v T^2} \frac{\partial T}{\partial t}}_{(III)}. \tag{1}$$

In Eq. (1), term (I) is the observed RH tendency, term (II) represents the contribution from $q$ change and term (III) from $T$ change. Term (III) includes the minus sign, which means that a positive value of this term indicates cooling and vice versa. For



the calculation of RH, $q$ and $T$ tendencies we use soundings released at 18:30 and 20:00 UTC for the Stable phase, at 20:00 and 23:00 UTC for the Jet phase, at 23:00 and 03:30 UTC for the Stable phase I, at 03:30 and 06:30 UTC for the Stratus phase II and at 06:30 and 11:00 UTC for convective phase. Other quantities in Eq. (1) are calculated as averages of all soundings within each phase. The results are shown in Fig. 7.

During the Stable phase, RH increases in the layer below 300 m a.g.l., with a maximum of about 4 % per hour (Fig. 7a). Above this level up to roughly 700 m a.g.l., RH is almost constant, while above RH decreases. The decrease of the temperature is mostly responsible for the increase of RH below 300 m, while on average there is a small positive moisture contribution. The median wind speed profile in this phase is less than 3 m s$^{-1}$ in the lowest 1 km. During the next phase, the layer with a significant increase of RH deepens to about 700 m a.g.l., with a maximum rate of 6 % per hour (Fig. 7b). The main bulk of
this change is caused by cooling, while moisture change is negligible during the Jet phase. The layer of the maximum change corresponds to the level of the NLLJ maximum. This is in agreement with results in Adler et al. (2018), who found that on average cooling is the main process leading to the increase of RH and saturation, while moistening contributes only little.

At the end of the Jet phase the clouds form, but are intermittent during the Stratus phase I, which is characterized by almost constant RH within the cloud layer, while below the cloud base a small decrease of RH of -1 % per hour is recorded (Fig. 7c).
Although we still observe cooling in the lowest 1 km during this phase, the rate is much lower compared to Stable and Jet phase. At the same time a competing, stronger negative contribution of the specific humidity change is observed. The median wind speed in the jet layer has even increased to 9 m s$^{-1}$, with the NLLJ maximum shifting upwards to the cloud top. As the LLC deck grows and becomes thicker, the NLLJ maximum shifts further upward towards the cloud top during the Stratus phase II (Fig. 7d). As this phase lasts even after sunrise, the daytime heating causes the weakening of the jet wind speed, and we
observe stronger temperature increase above the CTH than below the CBH. On average, a small positive RH tendency exists below the NLLJ maximum (about 0.5 % per hour), mostly due to positive $q$ tendency. In the Convective phase, the temperature continues to increase below 700 m a.g.l., and has a stronger contribution to negative RH tendency, while $q$ tendency is small (Fig. 7e).

### 4.3 Heat budget analysis

Since the contribution of the temperature change is more dominant, i. e. cooling is the dominant process for LLC formation, we investigate into more detail the heat budget during this IOP. The conservation equation for the mean potential temperature ($\theta$), with the molecular term neglected, is equal to (e.g. Garratt, 1992):

$$\underbrace{\frac{\partial \overline{\theta}}{\partial t}}_{(I)} = \underbrace{-\left(\overline{u}\frac{\partial \overline{\theta}}{\partial x} + \overline{v}\frac{\partial \overline{\theta}}{\partial y} + \overline{w}\frac{\partial \overline{\theta}}{\partial z}\right)}_{(II)} + \underbrace{\frac{1}{\rho c_p}\frac{\partial Q_j^*}{\partial z}}_{(III)} - \underbrace{\frac{L_v E}{\rho c_p}}_{(IV)} - \underbrace{\frac{\partial \overline{w'\theta'}}{\partial z}}_{(V)}, \tag{2}$$

where $c_p = 1004$ J kg$^{-1}$ K$^{-1}$ is the specific heat at constant air pressure, $\rho$ is density of the air, $Q_j^*$ is the net radiation
flux, $E$ is the mass of water vapor per unit volume per unit time being created by a phase change from liquid or solid to gaseous and $\overline{w'\theta'}$ is the kinematic heat flux. We use radiosoundings in the same manner as in the previous section to calculate the potential temperature tendency (term I). The advection term (II) is considered here as a residual term since we can not





calculate this term for each phase, but is estimated for a specific period in Sect. 4.3.1. The radiative flux divergence term (III) is determined using the radiative transfer (SBDART) model (Ricchiazzi et al., 1998). The latent heat release term (IV) is relevant in the case when clouds are present, and is determined from the LWP measurements, assuming that the liquid water content is linearly distributed over the cloud layer depth ($h$), therefore, the phase change term equals $\frac{L_v E}{\rho c_p} = -\frac{L_v}{c_p \rho h} \frac{\partial (LWP)}{\partial t}$. Finally,

the divergence of sensible heat flux (V) is determined using the mean surface values of sensible heat flux ($H_0$) and assuming linear decrease up to the top of the inversion layer (Stable phase), to the NLLJ maximum (Jet phase) or to the CBH during the nighttime conditions (Stratus phase I and II). For the daytime conditions, we analyzed measurements of turbulent fluxes obtained by unmanned arial system (UAS) ALADINA (Altstädter et al., 2015; Bärfuss et al., 2018) in order to get the insight into their characteristics. Since the flight times of the UAS do not correspond to the time period of the Convective phase, it is not possible to include them in the analysis directly. However, the analysis of 20 flights during the morning hours on 8 different days (not shown) indicates that it is reasonable to assume that sensible heat flux decreases linearly with height and equals $-0.2 H_0$ at the CTH.

Figure 8 shows the vertical profiles of heat budget terms for the five phases. The strong cooling of the layer below 300 m a.g.l. during the Stable phase leads to the formation of the stably stratified nocturnal ABL. A large part of the observed cooling is due to the surface longwave radiative flux divergence with a maximum cooling rate of $-0.22$ K per hour. When vertically averaged up to 275 m a.g.l., this term explains 29 % of the observed temperature change, while contribution from sensible heat flux divergence is only 7 %. The residual term is found to be the largest with 64 % contribution to the observed temperature change during this period (Fig. 8a). The large residual term is most likely caused by the cold pool outflow, which resulted from the early evening local rainfall event which occurred approximately 15 km south of Savè and moved westward during its life cycle (between 19:00 and 21:00 UTC) as revealed by X-band radar data (not shown). These results are in general agreement with findings by Sun et al. (2003), who found that the strongest radiative flux divergence is observed in the early evening under weak-wind and clear-sky conditions, which may contribute even up to 48 % to the observed cooling in the lowest 48 m a.g.l.

After the arrival of the NLLJ, the layer of the strongest cooling deepens to 700 m a.g.l., with the maximum cooling rate of $-1.2$ K per hour at the height of the NLLJ maximum (Fig. 8b). Below the NLLJ maximum, the contribution of longwave radiation, which is still active during this cloud-free period, and of sensible heat flux divergence to the observed cooling is approximately equal, i.e. 16 % and 12%, respectively. Based on $Ri$ and radial velocity variance values (Fig. 5a) there is evidence of increased turbulent mixing below the NLLJ maximum, suggesting upward turbulent transport of cold air leading to cooling and an increase of RH in this layer (Fig. 6a, b). However, the contribution of longwave radiation and sensible heat flux divergence (due to turbulent mixing) to the observed cooling below the NLLJ maximum is substantially lower compared to the 72 % contribution from the residual term. We assume that cooling due to the horizontal cold air advection, associated with the onset of NLLJ is most likely the reason for the strong decrease of temperature during the Jet phase and is considered in more detail in the next subsection. These results are in general agreement with results obtained for 11 different IOPs by Adler et al. (2018), i.e. the average 22 % radiative flux divergence contribution to the cooling during the Stable and Jet phase is in agreement with their results. Note that direct comparison of magnitudes is not advised since the methods applied differ





slightly between the studies, therefore, some differences in the contributions from sensible heat flux divergence and horizontal advection to the observed cooling are obtained during these two phases.

During the Stratus phase I, the observed cooling below the CBH is mostly due to the strong vertical wind shear, which causes an increase of the sensible heat flux divergence and it contributes to 48 % of the observed temperature change (Fig.

8c). The contribution of the radiative cooling is similar as in the Jet phase and is equal to 13 %. At the same time, the strong radiative cooling of $-1.3$ K per hour at the cloud top helps to maintain the cloud layer, which consequently evolves into dense stratus clouds by the end of this phase. The phase change term is positive within the cloud layer due to condensational heating. During Stratus phase II, the atmospheric conditions below roughly 500 m a.g.l. are quasi-stationary, therefore, no substantial difference between the different terms is observed. The most pronounced feature is the strong radiative cooling of $-3$ K per

hour at the cloud top (Fig. 8d). Normally, this strong radiative cooling at the cloud top leads to entrainment of air and increased turbulent mixing within the ABL (due to the density differences at the CTH) and, subsequently, to development of a strong capping inversion at the cloud top (e.g. Vilà-Guerau De Arellano et al., 2015). However, we do not observe the expected strong temperature inversion at the cloud top. Instead, the radiative cooling is counter-balanced by the large-scale advection of warmer air in the layer up to 2 km, leading to the observed weak heating at the cloud top. This horizontal warm-air advection

is accompanied by a wind direction change (Fig. 6).

After sunrise, solar radiation heats the surface, causing positive sensible heat flux and evolving CBL. Turbulent mixing due to buoyancy leads to upward transport of warm air from the surface and warming of the layer below the CBH, which explains 52 % of the warming of the CBL (Fig. 8e). Within the clouds, the phase change term is negative due to the evaporative cooling.

### 4.3.1   Horizontal temperature advection

The large residual during the Jet phase suggests that the horizontal cold air advection related to the NLLJ arrival has an important contribution to the observed temperature change, which consequently led to the saturation and LLC formation. In conditions of undisturbed south-westerly monsoon flow, Adler et al. (2017) and Deetz et al. (2018b) observed a frequent occurrence of a stationary front, which formed along the Guinean Coast in the afternoon and was located several tens of kilometers inland. This front was reflected in a strong gradient between the relatively cool air mass over the Gulf of Guinea

and warm air over land. Northward propagation of the front started after 16:00 UTC, i.e. after decay of turbulence in the CBL and it reached Savè region around 21:00 UTC. A similar stationary frontal structure was seen in the simulations by Grams et al. (2010) for the coast of Mauritania. They related its stationary during the day to a balance between horizontal advection within the onshore flow and turbulence in the CBL over the land. Based on previous numerical simulations, as well as the investigation of conditions along the coast (using radiosonde data) and at Savè, similar processes are expected to occur along

the Guinean Coast during the monsoon season. Specifically, we expect the horizontal cold advection to be related to the Gulf of Guinea maritime inflow which reaches Savè in the evening.

Figure 9(a, b) shows the vertical profiles of wind speed and potential temperature from radiosoundings at the coast (Accra) and Savè. The conditions at the coast at 17:00 UTC are characterized with strong monsoon flow of 8 m s$^{-1}$, compared to low winds at Savè. On the other hand, the conditions at Savè are much warmer, with a well mixed CBL. Generally, there are large





differences in the conditions between the coast and Savè, as the coastal station seems to be in the cold maritime air mass, while a well developed CBL dominates the conditions at Savè (Fig. 9a, b). At 23:00 UTC strong winds in Accra are still present, while the potential temperature decreased slightly. At the same time conditions at Savè have changed substantially: the wind profile is now characterized with a pronounced NLLJ up to 8 m s$^{-1}$, which is the same as at the coast, potential temperature

decreased considerably to about 299 K in the layer below 600 m a.g.l., i. e. it has nearly the same value as at the coast during the daytime. Based on these considerations we conclude that the front of the Atlantic Inflow with maritime air mass already passed Savè at this time. In the layer above 750 m a.g.l. conditions at Savè did not change considerably during this period.

The estimation of the horizontal temperature advection is based on several assumptions, which are described in detail in Adler et al. (2018). These include: (i) the assumption of homogeneous temperature distribution along the coast, (ii) neglecting

the zonal wind component, (iii) gradual (linear) increase of temperature in south-north direction within the maritime inflow air mass at a certain distance from the coast (due to the position of the maritime air mass front) and (iv) constant temperature in the continental ABL north of the front. Recent modeling studies (Adler et al., 2017; Deetz et al., 2018b) indicate that the maximum inland penetration of the maritime air mass front in the afternoon hours is between 50 and 125 km inland from the coast. Therefore, we estimate the contribution of horizontal advection to cooling at Savè during the Stable and Jet phases, using

radiosonde measurements at the coast (Accra) and Savè (Fig. 9c). The meridional temperature difference for four different front locations (50, 75, 100 and 125 km inland) is determined at 17:00 UTC, while the mean (meridional $v$) wind is the average of measurements at 17:00 UTC in Accra and 23:00 UTC at Savè. The horizontal advection ($-v\Delta\theta\Delta y^{-1}$) estimate indicates a maximum cooling rate of $-2.7$ K (6h)$^{-1}$ at the height of the NLLJ maximum. When we compare estimated horizontal advection to the potential temperature tendency at Savè for the time period 17:00 to 23:00 UTC, above the NLLJ maximum

there is an almost perfect fit (Fig. 9c), indicating that cooling in this layer can be explained by horizontal advection. Below the NLLJ maximum, the contribution of horizontal advection is 55 % to the observed cooling, which explains the large part of the heat budget residual term during the Jet phase and is in good agreement with the estimated average contribution from horizontal advection to cooling during the Stable and Jet phase for the whole DACCIWA campaign (Adler et al., 2018). Because for IOP 8 radiosonde data are available every 1.5 hours, we could determine the contribution of horizontal advection to the heat budget

for the Stable and Jet phase separately. Related to the Jet phase, which is characterized by the largest temperature change and the arrival of the NLLJ, the contribution by cold air advection accounts for 72 %, while its contribution accounts for 55 % when relating it to the period from 17:00 to 23:00 UTC.

## 5  Discussion

Satellite images reveal that during this particular IOP, first LLC form east of Atakora Mountains in Togo and upstream of

Oshogbo Hills in Nigeria. The location of LLC confined to mountainous regions suggests that orographically induced lifting is one important process relevant for their formation as found in modeling studies of Schuster et al. (2013) and Adler et al. (2017). In subsequent hours, the clouds in the region of Atakora Mountains extend towards the north-east, i. e. upstream of Savè. The evolution of LLC at Savè suggests that they are not advected from southwest, where they form first, but they most likely form



due to favorable atmospheric conditions, such as horizontal-cold air advection, longwave radiative cooling and sensible heat flux divergence. During this IOP, besides the main area inland where the first LLC form, they also form along the Guinea coast, a feature that was found in seasonal simulations and satellite observations (Schuster et al., 2013; van der Linden et al., 2015). However, in this case the clear spread farther inland can not be distinguished, instead, they seem to be quasi-stationary. During

the DACCIWA campaign, different regions of LLC formation were recorded, i.e. in some cases the first clouds formed over higher terrain, while for other nights they seem to be independent from terrain features and these are presented in more detail in Adler et al. (2018).

The heat budget analysis suggests that the most relevant process for the LLC formation is cooling due to horizontal cold air advection, which is associated with the arrival of maritime inflow indicated by cold air mass and southwesterly NLLJ.

Therefore, we consider the residual term of the heat budget during the Jet phase to represent the large scale horizontal temperature advection. Consequently, we compare this term with the horizontal temperature advection, which we determine using the 17:00 and 23:00 UTC radiosonde measurements from the coast and Savè. Although there are large uncertainties related to the estimation of the horizontal temperature advection (Adler et al., 2018), we are certain that this process contributes the most to the LLC formation. Our results suggest that the contribution from horizontal cold advection can be up to 72 % during the Jet

phase when cooling is the strongest. The difference in the horizontal advection estimation between this study and Adler et al. (2018) comes from the limitations in the temporal resolution of radiosondes, i.e. Adler et al. (2018) can not perform the heat budget analysis in a consistent manner since during some IOPs radiosondes between 17:00 and 21:00 UTC are not available. For this reason they need to include periods when horizontal advection is not active, which results with a lower contribution of 55 % during the longer time period (Stable and Jet phase). The horizontal cold air advection was also found to be an important

process in numerical simulation (Schuster et al., 2013; Adler et al., 2017). Our results highlight the importance of having a dense network of measurements, as well as resolving the atmospheric conditions at high temporal and spatial resolution in order to adequately quantify all processes relevant for LLC. For example, Schuster et al. (2013) quantified the cooling rate due to horizontal cold air advection to be $-5$ K $(12\mathrm{h})^{-1}$. This rate is similar to the observed potential temperature change between 17:00 and 23:00 UTC at Savè (Fig. 9c). However, as seen in this study the cooling due to the cold air advection at Savè oc-

curs during a much shorter period (20:00 to 00:00 UTC, Fig. 4), suggesting that our rate of $-1$ K h$^{-1}$ during the Jet phase gives a more reliable estimation of the horizontal advection magnitude. This cold air mass behind the maritime inflow is also drier compared to the continental air mass, thus giving an observational confirmation to findings from numerical simulations (Schuster et al., 2013; Adler et al., 2017).

On the other hand, previous observational studies identified upward mixing of moisture, due to the increased vertical wind

shear related to the NLLJ, as the main process for the cloud formation (Schrage and Fink, 2012). Although the vertical mixing of cold air from the surface layer aloft is found to have a non-negligible contribution to the overall cooling, which leads to the saturation and cloud formation, this is not the main process. Similar results are found by Adler et al. (2018), who assessed processes relevant for LLC formation on 11 IOPs during DACCIWA. The characteristics of NLLJ and its relation to LLC, i. e. the clouds form approximately three hours after the onset of NLLJ and their base corresponds to the NLLJ maximum height,





confirm the results of previous numerical studies. Additionally, our findings are in agreement with the results of Dione et al. (2018) who analyzed NLLJ-LLC relationship for the whole month of July during the DACCIWA campaign.

In the simulations of Schuster et al. (2013) and Adler et al. (2017), a shift of the NLLJ maximum towards the cloud top was found, which is now verified by the observations. This upward shift of the NLLJ maximum is found to be due to the

change in the stratification within the cloud layer, compared to the cloud-free period (Jet-to-Stratus phase I change). The wind shear induced mixing below the NLLJ maximum (at the height of 275 m a.g.l., Fig. 5a) reduces the gradient of the potential temperature, resulting in a less stably stratified layer below the NLLJ maximum than above. With the cloud formation, vertical gradients in wind speed and temperature decrease further, causing the thickening of the mixed layer, and consequently the shift of the inversion layer upwards. Adler et al. (2017) found the shift in stratification within the clouds to be caused by the

enhanced TKE, condensational heating, radiative cooling at the cloud top and upward motion in the stable ABL. All off the processes are observed in our study as well, except for the latter, since due to the uncertain measurements of vertical wind velocity by Doppler lidars we can not estimate the contribution of upward motion. Additionally, Adler et al. (2017) found an enhanced upward motion upstream of existing clouds to be the most relevant process for the evolution of dense extended clouds. Although this can not be estimated directly in this case study, there is evidence that this process contributes to the

formation of dense stratus during the Stratus phase II. Additionally, the study of Adler et al. (2018) shows that this mechanism occurs for other cases as well and it may explain the upstream expansion of LLC.

## 6   Summary and conclusions

The data collected during a comprehensive DACCIWA ground-based field campaign at the supersite in Savè (Benin) on 7–8 July 2016 are analyzed in order to investigate the diurnal cycle of LLC and related atmospheric processes. This particular

time period is chosen since the conditions during this case study reflect typical conditions and features related to undisturbed south-westerly monsoon flow. These typical features include the onset of NLLJ and the formation of LLC. The associated dynamic and thermodynamic conditions allow the definition of five different phases of the LLC diurnal cycle. These include: the Stable phase indicates the period after the sunset and before the onset of the NLLJ, when the wind speed in the ABL is low and increasing static stability causes the decoupling of the mixed layer from the stable ABL. The Jet phase starts with the

onset of NLLJ related to the arrival of Atlantic Inflow. The formation of LLC marks the beginning of the Stratus phase, which is divided in Stratus phase I, since the inhomogeneous cloud cover and unstationary atmospheric conditions are observed, and Stratus phase II, which corresponds to the period with a persistent stratus deck and quasi-stationary atmospheric conditions. The Convective phase starts approximately 1 hour after the sunrise and is associated with growing CBL and lifting of the cloud base.

Shortly after the sunset, the stably stratified nocturnal boundary layer developed due to the contributions of longwave radiative cooling of the ground (29 %), sensible heat flux divergence (7 %), as well as some local effects due to cold pool outflow from early evening convection, causing the decoupling of the mixed layer from the stable surface layer (Stable phase). Within the stable ABL, RH increased with the maximum rate of about 4 % per hour mostly due to strong cooling in this layer. A



beginning of the Jet phase is marked with the sudden onset of the southwesterly NLLJ, which is characterized with an increase of wind speed up to 8 m s$^{-1}$ and the NLLJ maximum height at about 275 m a.g.l. The strong wind shear below the NLLJ causes increased vertical turbulent mixing and consequently the erosion of the surface inversion, leading to the coupling of the residual and surface layer. The effect of horizontal cold air advection, related to the Gulf of Guinea maritime inflow, which

brings the cold maritime air mass and a prominent NLLJ wind profile, is found to have the dominant role on the observed strong cooling during Jet phase. The residual term of the heat budget is considered to correspond to horizontal temperature advection term and we find that it can contribute up to 72 % to the observed temperature change below the NLLJ maximum during the Jet phase. The contribution from radiative cooling and sensible heat flux divergence is 16 and 12 %, respectively. The cooling at a rate of $-1.2$ K per hour persists for approximately three hours causing the continuous increase of RH at a

rate of 6 % per hour, until finally the saturation is reached and LLC form at the height corresponding to the NLLJ maximum height. No significant contribution from the moistening is found during this phase. The unstationary conditions during the Stratus phase I are observed due to competing influences of processes leading to cooling, namely turbulent mixing and cold air advection, and a dry air advection, related to the drier maritime air mass behind the maritime inflow front, thus leading to inhomogeneous and thin cloud cover. A combining effect of the vertical wind shear below and above the NLLJ maximum and

the presence of LLC leads to the change in stratification, causing lower static stability in the sub-cloud layer and higher at the cloud top, which in turn results in an upward shift of the NLLJ maximum (from 275 to roughly 400 m a.g.l.). This shift of the NLLJ maximum towards the layer of maximum static stability continues in the Stratus phase II as quasi-stationary conditions are established. Turbulent mixing is an important factor leading to the cooling below the cloud base, while strong radiative cooling at the cloud top with a rate of approximately $-2$ K per hour helps to maintain thick stratus. In the morning, the 52 %

contribution from sensible heat flux divergence to the observed heating below the CBH is the largest and, consequently, leads to continuous warming of the CBL, lifting of the CBH and dissolution of LLC.

Overall, this study presents the first detailed observational analysis of the complete diurnal cycle of LLC and processes leading to their formation, maintenance and dissolution over southern West Africa. This comprehensive data set enabled the verification of the previous numerical results, as well as revealed some new findings enabling better understanding of processes

related to the West African monsoon. This mostly concerns the role of the Atlantic Inflow and associated horizontal advection of cold, but also drier air for the LLC formation. Furthermore, this detailed analysis of the diurnal cycle of LLC and related conditions and processes is expected to substantially contribute to the development of the conceptual model of the LLC life cycle.

*Data availability.*   After the DACCIWA embargo period, the data of Savè supersite will be available on the SEDOO database (Derrien et al.,

2016; Handwerker et al., 2016; Kohler et al., 2016; Wieser et al., 2016) for scientists interested in boundary-layer studies in southern West Africa.



*Competing interests.* The authors declare that they have no conflict of interest.

*Acknowledgements.* The DACCIWA project has received funding from the European Union Seventh Framework Programme (FP7/2007-2013) under grant agreement no. 603502. We thank the staff of KIT (Karlsruhe Institute of Technology) and UPS (University of Toulouse) for helping to install and run the equipment as well as to staff of INRAB in Savè for allowing to use their grounds for the experiment. We
5   thank Andreas Fink and his group for performing the radiosoundings at Accra and Christine Chiu for providing the SBDART code which uses Mie phase function calculations.





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





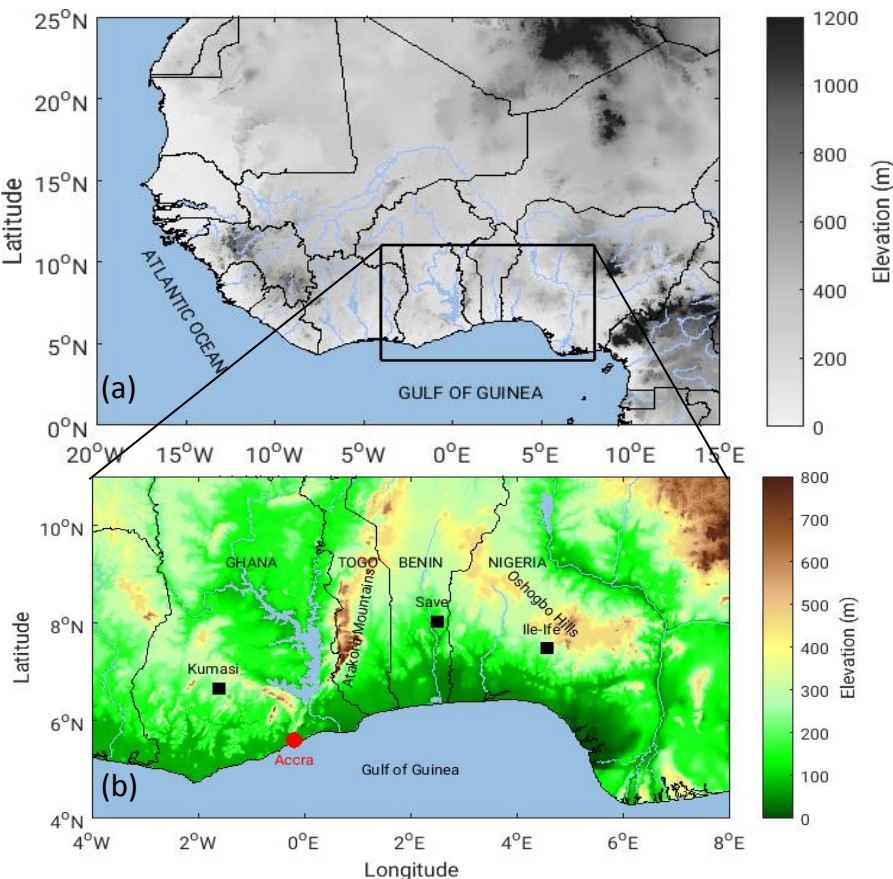

**Figure 1.** (*a*) The black rectangle indicates geographical location of the DACCIWA area of interest. (*b*) Topographic map of the investigation area. The three supersites, including Savè supersite, are indicated with black square. The coastal radiosonde station Accra is shown with red circle.







**Figure 2.** The difference in brightness temperature for spectral channels 10.8 (TIR) and 3.9 $\mu$m (MIR). The gray ares indicate TIR brightness temperature threshold at 283 K. The range of colorbar is indicated for each panel. Savè is indicated with black circle.



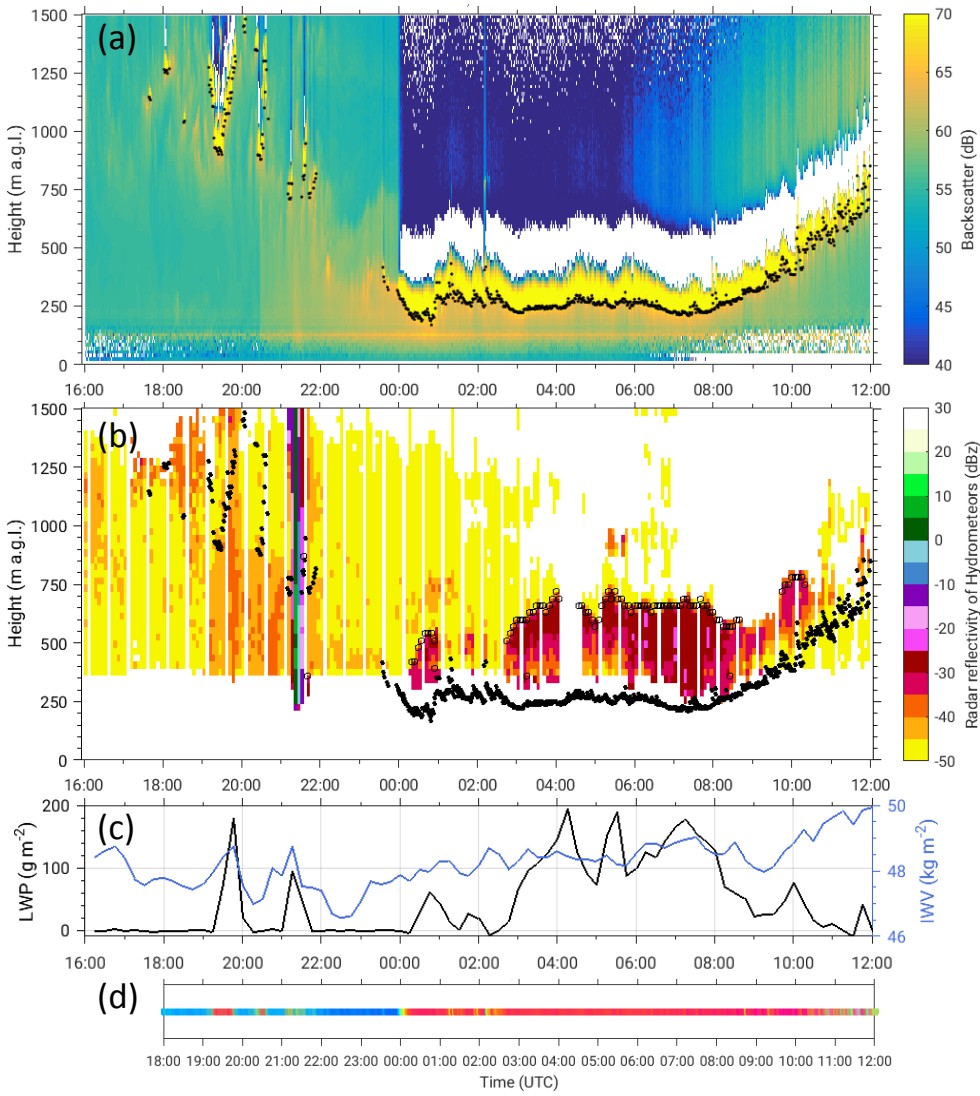

**Figure 3.** (*a*) Time series of ceilometer backscatter (color) and CBH (black dots) derived from the backscatter profiles. (*b*) The reflectivity of hydrometeors obtained by cloud radar (color), the CBH (black dots) and CTH (open circles) derived from the cloud radar using a threshold of -35 dBz. (*c*) Time series of 30-min averaged LWP and IWV from microwave radiometer. (*d*) The RGB image of sky conditions obtained by IR cloud camera.

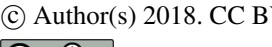



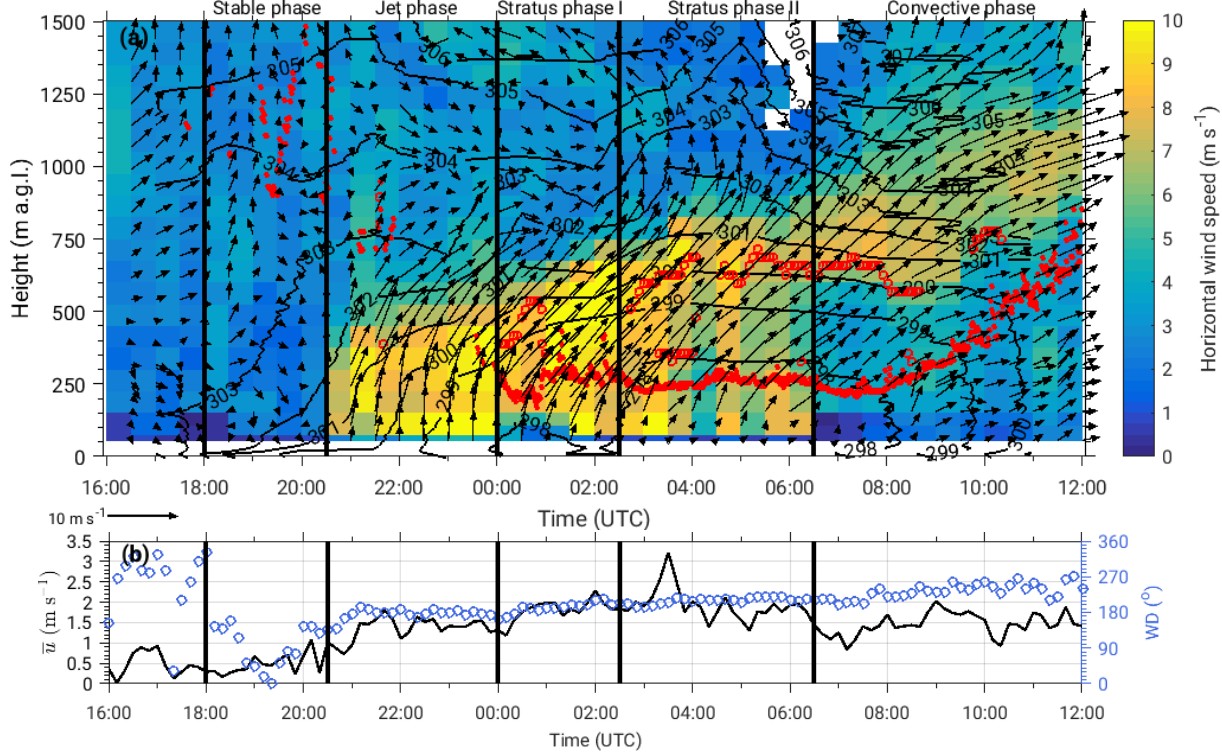

**Figure 4.** (*a*) Temporal evolution of the wind speed is shown in color, while arrows show wind direction obtained from UHF. Black contours show linearly interpolated potential temperature measured every 1.5 h by radiosondes. The red dots show CBH and CTH is shown with red circles. (*b*) Time series of the near-surface 10-min averaged wind speed (black) and wind direction (blue) measured by energy balance station. The vertical black lines indicate the beginning of five different phases observed during this IOP.





**Figure 5.** (*a*) The gradient Richardson number shown in color is calculated from radiosonde measurements. Black isolines show the vertical potential temperature gradient in K $(100 \text{ m})^{-1}$ calculated from radiosonde data. The white dots show CBH and white circles denote CTH. (*b*) Variance of the radial velocity obtained by lidar measurements. The black contours show horizontal wind speed (in m s$^{-1}$) measured by radiosondes, while black dots show CBH and black circles denote CTH. (*c*) Time series of the near-surface 30-min averaged TKE (black) and stability parameter ($\zeta$, blue) measured by energy balance station. The vertical black lines indicate the beginning of five different phases observed during this IOP.





**Figure 6.** Temporal evolution of relative humidity (color) and potential temperature in Kelvin (panel (*a*), isolines) and specific humidity in

g kg$^{-1}$ (panel (*b*), isolines) in the lowest 1.5 km obtained from radiosonde profiles performed every 1.5 h. The arrows show the horizontal

wind vector from radiosondes. The CBH is indicated with white dots and the CTH with open white circles. Time series of 10-min averaged

RH (*c*), temperature and specific humidity (*d*) measured by energy balance station 4 m a.g.l. The vertical red and black lines indicate the

beginning of five different phases observed during this IOP.





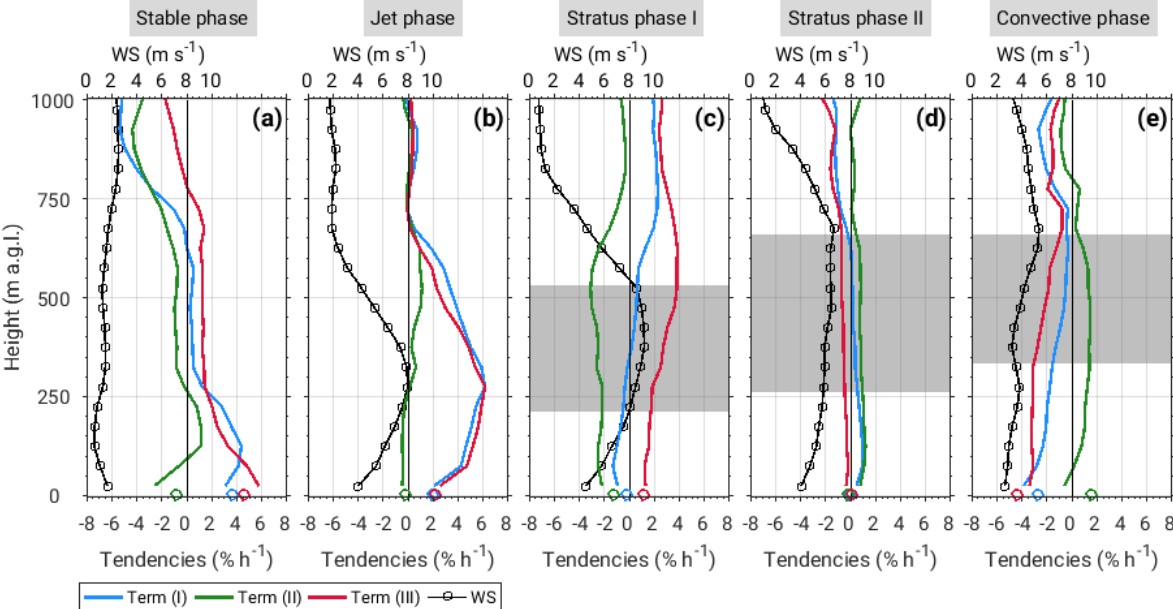

**Figure 7.** The observed RH tendency profiles obtained from radiosonde measurements for different phases during IOP 8 is shown in blue, while contributions from the temperature change term (Term I) and specific humidity term (Term II) are shown in red and green color, respectively. The circles show values of different terms of Eq. (1) obtained from surface measurements. The black circles denote the median horizontal wind profile for each phase. Shaded gray areas denote the mean cloud layer.





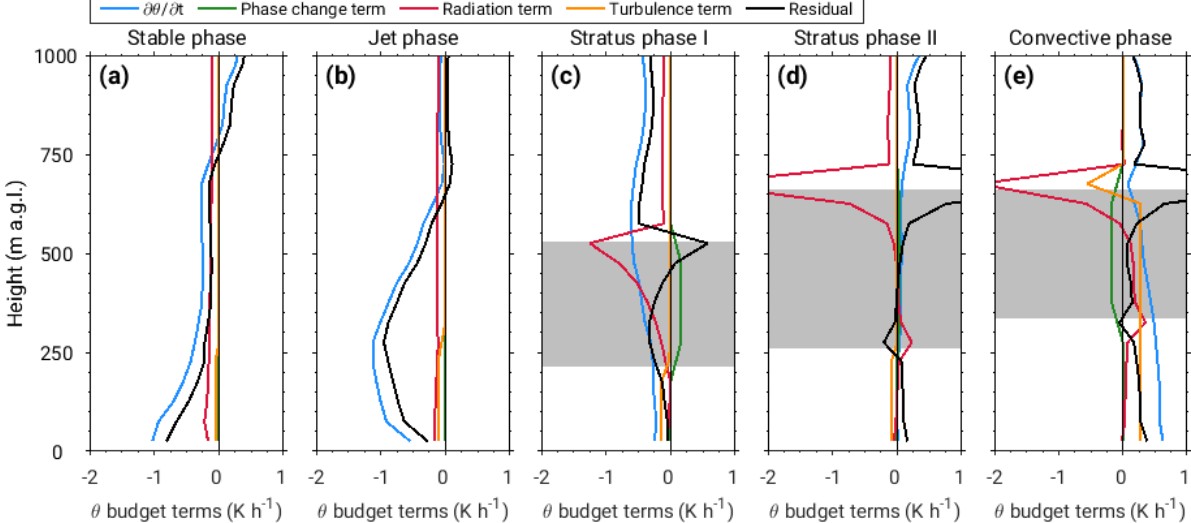

**Figure 8.** Vertical profiles of heat budget terms: potential temperature tendency (blue), phase change term (green), radiation flux divergence term (red), divergence of the sensible heat flux (orange) and residual term (black) shown for different phases during IOP 8. The shaded gray area indicates the mean cloud layer.





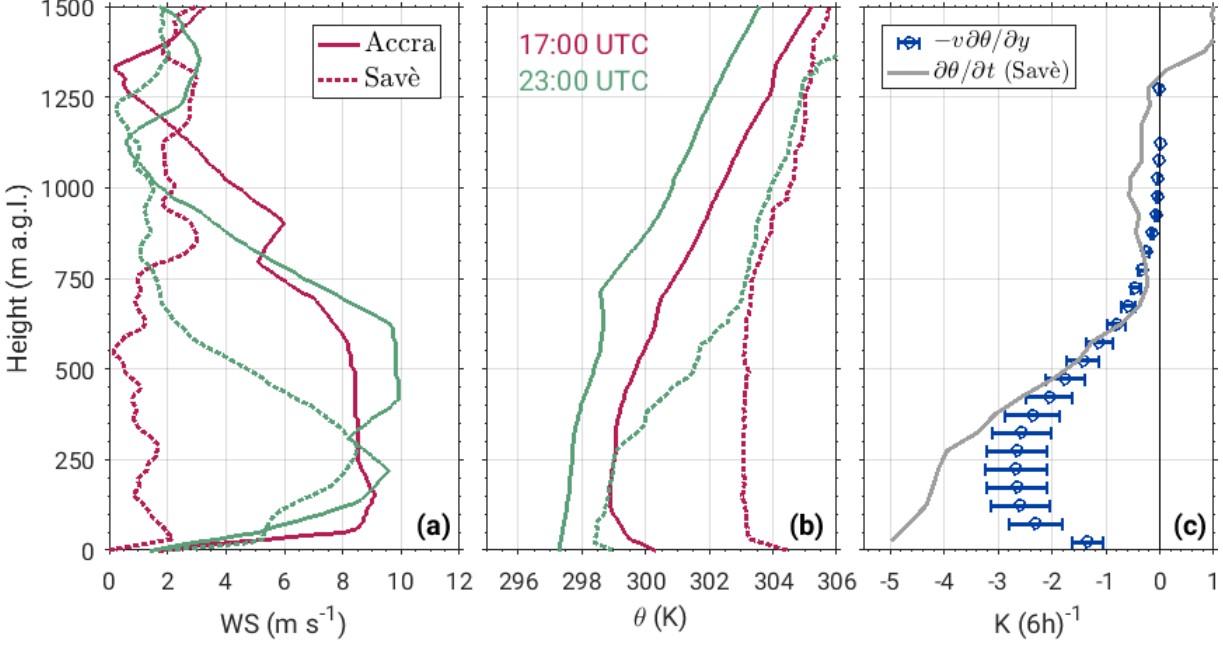

**Figure 9.** Comparison of vertical profiles of horizontal wind speed (*a*) and potential temperature (*b*) in Accra (solid line) and Savè (dashed line) at 17:00 and 23:00 UTC (shown in red and green color, respectively.) (*c*) The mean horizontal advection estimated between the coast and Savè for different front locations (50, 75, 100 and 125 km from the coast) during 6 h period (17:00-23:00 UTC) using radiosonde measurements in Accra and Savè (blue), while error bar show one standard deviation. The potential temperature change observed at Savè during this period is shown in gray.