# Peer review of "The observed diurnal cycle of low-level stratus clouds over southern West Africa: a case study"

_Atmospheric Chemistry and Physics, 2018_

## Referee Comment (RC1) · Anonymous Referee #1 · 9 Sep 2018

The study focuses on a diurnal variation of low level clouds (LLC). The analyses are well presented with field observations. The authors concluded that temperature advection played the major role on the formation of the LLC.

Although the authors briefly mentioned the possible explanation of the LLC formation as a result of vertical turbulent mixing, I would still challenge the authors to think more about the role of turbulent mixing in the growth of the LLC.

Because of difficulties of calculating temperature advection, the authors relied on estimating temperature advection as a residual term with several assumptions. The temperature decrease around 2100 UTC could be due to the vertical transport of cold air

from the radiatively cooled surface by turbulent mixing associated with the increase of wind speed especially considering of non-local turbulence eddies (Sun et al. 2016, BLM), which is clearly demonstrated in Fig.4. That is, as the surface was radiatively cooled, the cold air adjacent to the surface was generated. Once wind speed increased, the cold air was transported upward by the enhanced vertical mixing. As a result, the air below the NLLJ was relatively well mixed, which is shown in the vertical temperature profiles and the decrease of the specific humidity, q, at 4 m (Fig. 6). The decreased air temperature led to the increase of RH, the formation of the LLC, and the thickening of the LLC. The evolution of the NLLJ should be associated with the vertical stable stratification of the air below. The increase of the wind speed could not be fast at the beginning of the NLLJ formation due to the stable stratification. As the stable stratification was reduced by the vertical turbulent mixing, the air layer below the NLLJ approached near neutral, and the downward sensible heat flux would be relatively small. Then, the air temperature would remain stationary as the cold air cannot be generated at the surface fast enough especially with the formation of the LLC, which is indeed evident in Fig. 4a. The timing of the sharp temperature decrease and the wind speed increase around 2100 UTC seems to be an indication of the role of the vertical turbulent mixing. The role of vertical mixing could also be seem in vertical variations of specific humidity, which are not shown in the manuscript but should be similar to vertical variations of potential temperature. If the advection plays a significant role in the LLC formation, I would expect a continuous increase of specific humidity after 2100 UTC as advection would bring in a different air mass into the study site. The initial small increase of specific humidity at 4 m around 2100 UTC could be due to the vertical transport of water vapor from the air layer below 4 m, and the steady decrease of q with time in Fig. 6d also suggests the important role of vertical mixing as the high q air was transported upward and the low q air was transported downward.

I am not saying that the temperature advection didn't play any role on the LLC formation. However, vertical turbulent mixing associated with the wind speed increase could play a major role in the LLC formation based on the observations presented in the

manuscript. A close look at the timing of all the observations may help to determine different roles of horizontal advection and vertical turbulent mixing.

Minor comments:

What is the surface type? Any observation evidence for a constant albedo? What is the vertical resolution of the microwave radiometer? Is the vertical divergent of radiative fluxes sensible to the vertical distributions of liquid water and water vapor? Figure 2. It will be helpful to add explanation of different color bars for nighttime and daytime, and why the nighttime technique for detecting the cloud cover cannot be used for daytime. L. 19. "...without a clear cloud radar signal." Is this an instrument issue or no clouds?
* * *

---

## Referee Comment (RC2) · Anonymous Referee #2 · 18 Oct 2018

Review of the article titled "The observed diurnal cycle of nocturnal low-level stratus clouds over southern west Africa: a case study" by Babic and coauthors for publication in the journal atmospheric chemistry and physics. The authors have used data collected during the DACCIWA field campaign during a single day to understand the causes for the presence of low-level clouds. They have used radiosonde data to do budgets of relative humidity and heat. The main conclusion from the study is that the advection of colder air from the ocean to the site to lead to the formation of the clouds. The article is relatively straight-forward to understand and the authors have clearly described the data and methods used in the study. The overall scientific novelty of the study is however unclear. As they have analyzed a case with low level wind jet with a

southerly flow, it is apparent that it will have large warm and moist advection from the ocean. Hence, I don't think the main conclusion that advection is important for the formation of low-level clouds is novel. I recommend this article for major revisions. Major Comments The title is confusing as it has diurnal cycle and nocturnal in it. The two words sort of contradict each other. Maybe you can use something like "A case-study of the nocturnal low level stratus clouds over the West Africa from the DACCIWA field campaign." This is a mere suggestion, please feel free to use something else. The main concern I have is that that the conclusions are solely based on centered difference taken from the radiosonde data. This makes the study weak, as there are no uncertainty estimates and also no verification from other variables. To get around this issue, maybe you can i) show the ECMWF model reported largescale temperature, moisture and winds in the study area, and/or ii) propagate the uncertainty in all of the variables in equation i) and equation ii) to show some variability in the terms shown in Figure 7, 8 and 9. The way you have setup SBDART, there will be large uncertainty in the radiative fluxes. Also, please mention the assumed cloud droplet effective radius. The assumed profiles of sensible heat fluxes also make a huge difference in the calculations. It is not clear why the authors have chosen to use different profiles of sensible flux for different atmospheric phases that are only few hours apart. Section 4.2: you have calculated the RH budget to understand whether q or T has greater impact on the RH. I agree with you regarding the premise that a moisture advection can happen but it wouldn't necessarily lead to saturation and clouds, however I disagree the way you have gone about it. The (II) and (III) terms in equation i) have the tendency terms of water vapor mixing ratio and temperature in them. Now as you have shown in the section 4.2, the tendency of temperature also depends on the advection. So I recommend you not to use the basic Clausius-Clapeyron equation, but do a classical moisture budget assuming a well-mixed boundary layer. See Caldwell et al. (2005 JAS) or Kalmus et al. (2014 J. Climate). This will enable to understand if the changes in the moisture are locally generated or a result of large-scale advection. In the same vein, it will be great if you can show the sensible and latent heat fluxes during the study period. Thanks.

[Figure]

It will be great if the authors can calculate the lower tropospheric stability of the study area including for the soundings launched in Accra. The last two sections (5 and 6) are unnecessarily long and do not add any value to the manuscript. They have several repetitions and I think could be severely shortened and merged together. Thanks.

Minor Comments Page 2, Line 8: I think you should add the "in that area" to this sentence. Otherwise the sentence is very generic. This doesn't apply to all of the LLC. Page 2, Line 32: "processes that" instead of "processes which". Section 2.2: Please mention the frequency, temporal resolution and range resolution of the cloud radar. Same for the ceilometer and IR camera. Page 6, Line 22: Please remove the word "apparently" Page 9, line 9: "deck" not "decked". Figure 1: Please make the distance on the x- and the y-axis the same. Currently the aspect ratio is not one. Figure 2: The caption needs to mention what is shown in all panels. Currently it is not clear what is shown. Figure 3: Please change the color-scale of panel (b) from -50 to 10 dBz. It will be nice if you can clean up the data to only show returns from hydrometeors. The SNR can do that. It will be also nice if you can zoom-in the panel (d) and show a color-scale for the panel (d). thanks. It will be nice if the authors also put this study in perspective of those done during the RADAGAST campaign (Miller and Sling, 2007 BAMS; Collow et al. 2016 QJRMS etc.)

---

## Author Comment (AC1) · 14 Nov 2018

Please find our response to the comments in the attached .pdf file.

Please also note the supplement to this comment:
https://www.atmos-chem-phys-discuss.net/acp-2018-776/acp-2018-776-AC1-supplement.pdf

---

## Author Response (AR1)

Dear Co-Editor,

We are thankful for the valuable suggestions and comments by the referees. Please find below our point-by-point responses to the referees' comments. The review comments are shown in black, while our responses to individual comments are in blue. In the revised manuscript new or changed text is highlighted as red.

Sincerely,

Karmen Babić on behalf of all coauthors

**Referee 1:**

We would like to thank Referee #1 for the thorough comments and suggestions. Detailed response to each comment is provided below.

The study focuses on a diurnal variation of low level clouds (LLC). The analyses are well presented with field observations. The authors concluded that temperature advection played the major role on the formation of the LLC. Although the authors briefly mentioned the possible explanation of the LLC formation as a result of vertical turbulent mixing, I would still challenge the authors to think more about the role of turbulent mixing in the growth of the LLC. Because of difficulties of calculating temperature advection, the authors relied on estimating temperature advection as a residual term with several assumptions. The temperature decrease around 2100 UTC could be due to the vertical transport of cold air from the radiatively cooled surface by turbulent mixing associated with the increase of wind speed especially considering of non-local turbulence eddies (Sun et al. 2016, BLM), which is clearly demonstrated in Fig.4. That is, as the surface was radiatively cooled, the cold air adjacent to the surface was generated. Once wind speed increased, the cold air was transported upward by the enhanced vertical mixing. As a result, the air below the NLLJ was relatively well mixed, which is shown in the vertical temperature profiles and the decrease of the specific humidity, q, at 4 m (Fig. 6).

We completely agree with the referee. Indeed, the timing of the NLLJ onset and, consequently, increased turbulent mixing is most likely the reason for the strong decrease of temperature observed up to 800 m.g.l. in the first part of the Jet phase (Fig. 1) due to the vertical transport of cold air (Figs. 4 and 5 in the paper). The discussion related to this is now introduced on Pg. 8, lines 29–31.

The decreased air temperature led to the increase of RH, the formation of the LLC, and the thickening of the LLC. The evolution of the NLLJ should be associated with the vertical stable stratification of the air below. The increase of the wind speed could not be fast at the beginning of the NLLJ formation due to the stable stratification. As the stable stratification was reduced by the vertical turbulent mixing, the air layer below the NLLJ approached near neutral, and the downward sensible heat flux would be relatively small. Then, the air temperature would remain stationary as the cold air cannot be generated at the surface fast enough especially with the formation of the LLC, which is indeed evident in Fig. 4a. The timing of the sharp temperature decrease and the wind speed increase around 2100 UTC seems to be an indication of the role of the vertical turbulent mixing.

Yes, we agree with the referee. As evident from Fig. 1, the evolution of the NLLJ was related to the static stability of the atmosphere, with the jet axis just above the inversion layer at 275

[Figure]

**Figure 1:** The vertical profiles of wind speed and potential temperature measured by radiosondes during Stable (18:29 and 20:00 UTC) and Jet phase (21:28 and 23:00 UTC).

m a.g.l. The maximum wind speed in the NLLJ is very similar during the Jet phase where static stability is decreased (due to the vertical mixing) compared to Stable phase, but is still not near neutral (21:28 and 23:00 UTC profiles). The NLLJ wind speed increases in the Stratus phase I (Fig. 2). Once LLC have formed, they also affect the evolution of the NLLJ by shifting its axis towards the cloud top due to the change in stratification caused by clouds, which is evident from Fig. 2. This is discussed in the paper on Pg. 15, lines 5–15. Indeed, the potential temperature profiles measured during the Stable phase II (03:29, 04:59 and 09:30 UTC, 2), when thick and homogeneous LLC exist, indicate stationary conditions.

The role of vertical mixing could also be seen in vertical variations of specific humidity, which are not shown in the manuscript but should be similar to vertical variations of potential temperature. If the advection plays a significant role in the LLC formation, I would expect a continuous increase of specific humidity after 2100 UTC as advection would bring in a different air mass into the study site. The initial small increase of specific humidity at 4 m around 2100 UTC could be due to the vertical transport of water vapor from the air layer below 4 m, and the steady decrease of q with time in Fig. 6d also suggests the important role of vertical mixing as the high q air was transported upward and the low q air was transported downward.

The vertical variations of specific humidity are shown in Fig. 6b in the paper, and additionally, individual vertical profiles are shown in Fig. 2. We observe an increase in specific humidity at 21:30 UTC, which occurs simultaneously with the increase of turbulent mixing. We do agree with the referee that this increase is most likely due to the vertical transport related to NLLJ. This is now included in the revised version of manuscript on Pg. 9, lines 10–13. However, measurements at 23:00 UTC and later, show a decrease of specific humidity below 700 m a.g.l. This continuous decrease of specific humidity (after 23:00 UTC), despite increased turbulent mixing below the NLLJ axis and cloud base, indicates that this is a signature of the different air mass related to the arrival of the Gulf of Guinea maritime inflow, which is accompanied by the NLLJ. Contrary to common expectation that the maritime air mass will bring relatively moister air mass, we find exactly the opposite: the maritime air mass is relatively drier and cooler than

[Figure]

**Figure 2:** The vertical profiles of wind speed, potential temperature and specific humidity measured by radiosondes during IOP 8.

the environment further inland, which is represented by measurements at Savè site. We assume that this is due to the high evapotranspiration over the land and lower evapotranspiration over the cold ocean.

I am not saying that the temperature advection didn't play any role on the LLC formation. However, vertical turbulent mixing associated with the wind speed increase could play a major role in the LLC formation based on the observations presented in the manuscript. A close look at the timing of all the observations may help to determine different roles of horizontal advection and vertical turbulent mixing.

During the Jet phase the observed change of temperature in the layer from the surface up to NLLJ axis height (275 m a.g.l.) is −0.58 K per hour. If this whole change was only due to the

turbulent mixing associated with the NLLJ, the surface sensible heat flux during this phase should be roughly $-50$ W m$^{-2}$ in order to cool this layer. The measured mean sensible heat flux during this phase was -13.2 W m$^{-2}$, which can contribute to temperature decrease of $-0.14$ K per hour. In our case, the onset of NLLJ was accompanied by the arrival of maritime inflow, a cool (and drier) air mass propagating northwards from the coast in the late afternoon and the evening. The observed temperature change can not be explained by radiative cooling and vertical mixing alone, but is evidently related to the horizontal advection of this cold maritime air mass. This is also evident from the analysis of other IOPs (Adler et al. 2018).

**Minor comments:**

- What is the surface type?
  The surface type at Savè supersite is grass and bushes. This information is included on Pg. 3, line 33.

- Any observation evidence for a constant albedo?
  Our observations indicate that albedo is constant during the daytime (between 07:00 and 16:00 UTC) as shown in Fig. 3 for 7th and 8th July 2018. Slight increase is observed after 16:00 UTC, when the angle of direct solar radiation becomes low. The mean albedo is 0.2. This information is used as the input for the radiative transfer SBDART model (Pg. 6, line 1).

[Figure]

**Figure 3:** The albedo timeseries measured on 7th and 8th July 2016.

- What is the vertical resolution of the microwave radiometer?
  During the DACCIWA campaign a passive microwave radiometer (HATPRO, Humidity and Temperature Profiler) designed by Radiometer Physics Gmbh was used. The instrument measures sky brightness temperature at 14 frequencies (7 frequencies between 22 und 31 GHz (K-band) along the wing of the 22.235 GHz rational water-vapour line und 7 frequencies between 51 und 58 GHz (V-Band) along the wing oft the 60 GHz oxygen absorption complex). From these measurements temperature and humidity profiles as well as the integrated liquid water path (LWP) and the integrated water vapor (IWV) content are obtained. While IWV has an accuracy of less than 1 kg m$^{-2}$ (Pospichal and Crewell 2007), the vertical resolution of humidity is low as only two of the seven available water vapor channels are independent (Löhnert et al. 2009). Adding the information from 9 elevation scans performed at low angles to the standard zenith observations for the opaque

oxygen complex allows to obtain temperature profiles with a higher spatial resolution and an accuracy of less than 1 K below around 1.5 km (Crewell and Löhnert 2007). Therefore, the vertical resolution of temperature profiles is 50 m in the boundary-layer mode in the range 0-1200 m a.g.l. and 200 m up to 5000 m a.g.l. and 400 m between 5000 and 10000 m a.g.l. The vertical resolution of humidity measurements is slightly coarser, i.e. 200 m up to 2000 m a.g.l., 400 m and 800 m up to 5000 m and 10000 m a.g.l., respectively. In our paper we only use information on LWP and IWV, which are integrated values through the whole atmosphere and do not have a vertical resolution.

- Is the vertical divergent of radiative fluxes sensible to the vertical distributions of liquid water and water vapor?

  The vertical divergence of radiative fluxes is determined using the radiative transfer model SBDART. In this model only the information on integrated liquid water path is used. The LWP of a cloud is specified in units of g m$^{-2}$. This is another way to specify cloud optical depth (Ricchiazzi et al. 1998). The calculation of radiative fluxes is not very sensitive to input values of the LWP during the nighttime and early morning, but shows more pronounced sensitivity when LLC start to break up as seen from Fig. 4.

[Figure]

**Figure 4:** The sensitivity of cooling rates ($\partial\theta/\partial t = -\frac{1}{g\rho}\frac{\partial Q^*}{\partial z}$) calculated for different input values of liquid water path and aerosol optical depth for different times of radiosonde releases during IOP 8.

However, since we have used measured values of LWP, this reduces the uncertainty of calculated radiative flux profiles (Fig. 5). Additionally, the vertical profile of water vapor is determined from radiosonde measurements of relative humidity and temperature. These profiles do not show significant vertical variability, especially in the lowest 1500 m a.g.l., but decrease linearly with height. Besides, no significant difference between different profiles is observed.

- Figure 2. It will be helpful to add explanation of different color bars for nighttime and

[Figure]

**Figure 5:** The sensitivity of net longwave radiation calculated for different input values of LWP and AOD for different times of radiosonde releases during IOP 8. Black circle indicates near-surface measurements of net radiation from energy balance station.

daytime, and why the nighttime technique for detecting the cloud cover cannot be used for daytime.

At night, the difference between the 10.8 and the 3.9 $\mu$m brightness temperatures is a proxy for cloud droplet size and can thus be used to detect low clouds (smaller droplets) (Hunt 1973). However, during daytime, the channel at 3.9 $\mu$m measures a mixture of outgoing thermal and reflected solar radiation, so that this method does not work after sunrise (Cermak and Bendix 2008). As such, we have used information from the visible channel during daytime. To make this switch in techniques obvious and transparent for the reader, different color bars are chosen for the different techniques. This explanation is now introduced on Pg. 5, lines 10–19 (highlighted in red).

- L. 19. ". . .without a clear cloud radar signal." Is this an instrument issue or no clouds? In this case the missing cloud radar signal is due to both, cloud properties and instrument performances/limitations. In the period between 01:00 and 03:00 UTC an inhomogeneous cloud cover, with patchy and thin LLC is observed. Since the received power of cloud radar depends on system properties, atmospheric attenuation and on the backscatter cross section and the backscattered energy is proportional to the sixth power of target diameter (Banta et al. 2013), the observed low reflectivity indicates very small particles within the cloud.

**Referee 2:**

We would like to thank Referee #2 for the thorough comments and suggestions. Detailed response to each comment is provided below.

Review of the article titled "The observed diurnal cycle of nocturnal low-level stratus clouds over southern west Africa: a case study" by Babic and coauthors for publication in the journal atmospheric chemistry and physics. The authors have used data collected during the

DACCIWA field campaign during a single day to understand the causes for the presence of low-level clouds. They have used radiosonde data to do budgets of relative humidity and heat. The main conclusion from the study is that the advection of colder air from the ocean to the site to lead to the formation of the clouds. The article is relatively straight-forward to understand and the authors have clearly described the data and methods used in the study. The overall scientific novelty of the study is however unclear. As they have analyzed a case with low level wind jet with a southerly flow, it is apparent that it will have large warm and moist advection from the ocean. Hence, I don't think the main conclusion that advection is important for the formation of low-level clouds is novel. I recommend this article for major revisions.

Although it might seem that the main conclusion of our study is regarding the role of horizontal cold air advection for the LLC formation, this is not the most important finding of our study. This study is the first observational study of the complete life cycle of LLC and associated atmospheric conditions and processes in the region of southern West Africa. It provides the first observational evidence of the processes involved in the formation, maintenance and dissolution of LLC. Namely, it shows that cooling is the most dominant contribution to the increase of relative humidity and, consequently, saturation. It also provides observational evidence that the air mass related to the Gulf of Guinea maritime inflow is relatively cooler and drier than the air mass over land. This is an important result highlightning the difference in atmospheric conditions over southern West Africa and in the areas to the north in the Sahel region, where NLLJ is considered to bring cool and moist air mass.

**Major Comments:**

- The title is confusing as it has diurnal cycle and nocturnal in it. The two words sort of contradict each other. Maybe you can use something like "A case-study of the nocturnal low level stratus clouds over the West Africa from the DACCIWA field campaign." This is a mere suggestion, please feel free to use something else.

  Indeed, we realize that the current title can be confusing, thank you for pointing this out. This is probably the first observational study which shows the complete life cycle of LLC and associated atmospheric conditions and processes, from formation until their break up, and therefore we want to keep "diurnal cycle" in our title. Since "nocturnal" refers to the part of the day when LLC form, we decide to leave this out and will have a new title: The observed diurnal cycle of low-level stratus clouds over southern West Africa: a case study.

- The main concern I have is that that the conclusions are solely based on centered difference taken from the radiosonde data. This makes the study weak, as there are no uncertainty estimates and also no verification from other variables. To get around this issue, maybe you can i) show the ECMWF model reported large scale temperature, moisture and winds in the study area, and/or ii) propagate the uncertainty in all of the variables in equation i) and equation ii) to show some variability in the terms shown in Figure 7, 8 and 9.

  Each of the radiosondes released during the campaign was calibrated in the same chamber, so this means that there could be an offset due to the conditions in the calibration chamber. However, this will only affect the absolute error, which is the same for all sondes. The M10

radiosondes from Meteomodem were used to obtain vertical profiles of different parameters. For this system, temperature measurements have uncertainty (reproducibility) of 0.2 $^oC$ and the uncertainity of relative humidity measurements is 2 %. The time derivatives in Eqs. (1) and (2) are approximated with finite forward difference using the radiosondes which correspond to the beginning and the end of the phase. All other terms in Eq. (1) are calculated based on the mean values for the individual phase using all available radiosonde measurements within that particular phase. The conditions in West Africa are generally not well represented in standard NWP or climate models (Knippertz et al. 2011; Hannak et al. 2017). This is one of the reasons why observations are preferred for this region. The data set collected during the DACCIWA field campaign is unique and so far the most comprehensive data set of high-quality measurements of the meteorological conditions in this region. Due to the limitations that numerical models of coarse resolution have in representing the state of the boundary layer, the model output would only, if at all, be useful to analyze the large scale conditions. Therefore, we followed on referee's suggestion and calculated the range of uncertainty for each of the terms in Eq. (1) by calculating the propagation of uncertainty based on the known uncertainties of radiosonde measurements, which are $\sigma_T = 0.2$ $^oC$ for $T$ and $\sigma_{RH} = 2$ % for RH. This is shown in new Fig. 7 and discussed on Pg. 10, lines 12–13 and 17–18. Unfortunately, it is not possible to use the error propagation calculation in a consistent manner for calculating uncertainties of the heat budget terms. The different terms in Eq. (2) are determined using different data and methods, and only for some of them measurement uncertainty is available. For example, the uncertainty range can be estimated for potential temperature tendency and it is consistent with results now shown for term (III) of Eq. (1). Measurement performances of HATPRO when measuring LWP are very high, with RMS of $< 2$ g m$^{-2}$. Therefore, the uncertainty range of phase change term is also very small. It is shown above that SBDART is not very sensitive to input parameters which were not measured directly and, therefore, the uncertainty of this term is also small during the nighttime. The uncertainty of the sensible heat flux divergence increases in case when sensible heat flux varies significantly during the particular phase, which in our case is during Convective phase only. Considering all this and the fact that Fig. 8 is already quite busy, we do not include uncertainty range for each of the terms shown. The information on the range of variability is provided in accompanying paper by Adler et al. (2018). In the revised version of the manuscript discussion on the range of uncertainty for the sensible heat flux divergence is provided on Pg. 12, lines 32–34. Also, error bars for the estimation of horizontal advection term are already included in Fig. 9.

- The way you have setup SBDART, there will be large uncertainty in the radiative fluxes. Also, please mention the assumed cloud droplet effective radius.
  We believe that the way we have set up SBDART using as many as possible observational data as input parameters, reduces the unceratinity of calculated radiative fluxes. Besides this, we preformed sensitivity test for those parameters which were not measured, such as sensitivity test for the boundary layer aerosols (BLA), visibility, but also sensitivity tests

for measured parameters, such as, liquid water path (LWP) and aerosol optical depth (AOD) shown in Figs. 4 and 5. Our tests show that the largest uncertainty is in the modeled shortwave (SW) incoming ($SWD$) and outgoing ($SWU$) radiation for the time period after 09:00 UTC, when break up and lifting of LLC start (however, our main focus is not on this late morning period). These components show sensitivity for input of BLA, LWP and AOD (Figs. 6, 7, 8 and 9), but not to visibility (Fig. 10). The cooling rates for different visibility input are shown in Fig. 10.

[Figure]

**Figure 6:** The sensitivity of the radiative flux cooling rate, net radiation ($Q^*$), net SW ($SW_{net}$) and longwave ($LW_{net}$) radiation and SW and LW flux components ($SWD$, $SWU$, $LWD$ and $LWU$) calculated for different input values of BLA and AOD for 06:30 UTC radiosonde release during IOP 8. Black circle indicates near-surface measurements of net longwave radiation from energy balance station.

[Figure]

**Figure 7:** The same as in Fig. 6 but for 08:00 UTC profile.

The modeled longwave (LW) flux components are not sensitive to input profiles of BLA, visibility, AOD (Figs. 6, 7, 8 and 9) or LWP during the nighttime (Fig. 11), which means

[Figure]

**Figure 8:** The same as in Fig. 6 but for 09:30 UTC profile.

[Figure]

**Figure 9:** The same as in Fig. 6 but for 11:00 UTC profile.

that the uncertainty in calculated cooling rates (i.e. vertical flux divergence) during the nighttime is very small. The range of variability is very small even for different atmospheric conditions as seen from Adler et al. (2018, their Fig. 10). Although LW flux components show sensitivity to LWP input values after 09:00 UTC (Fig. 11), their vertical variability is similar, which results in negligible differences in the calculated vertical divergence of radiative flux. Finally, we chose the configuration which gave the best agreement of net LW and net SW radiation with the near-surface observations (Pg. 6, lines 8–10). We use the default value of cloud droplet effective radius of 8 $\mu$m as this value is within the range of aircraft measurements in the area (Deetz et al. 2018; Taylor et al. 2018), indicated on Pg. 6, line 6.

[Figure]

**Figure 10:** The sensitivity of the radiative flux cooling rate for different input values of visibility for different times of radiosonde releases during IOP 8.

[Figure]

**Figure 11:** The sensitivity of net LW radiation calculated for different input values of LWP and AOD for different times of radiosonde releases during IOP 8. Black circle indicates near-surface measurements of net longwave radiation from energy balance station. For 11:00 UTC each panel shows profiles for different AOD values $= 0, 0.358, 0.532$ and $0.973$, from left to right.

- The assumed profiles of sensible heat fluxes also make a huge difference in the calculations. It is not clear why the authors have chosen to use different profiles of sensible flux for different atmospheric phases that are only few hours apart.

  For all five phases the same linear profile of sensible heat flux is assumed using the observed value from the energy balance station as surface conditions. This assumption is based on a well known profile from previous research (e.g., Stull 1988, Figs. 3.1, 3.2, 3.3, 4.19, ) as well as based on unmanned aerial vehicle measurements. As these phases are characterized with different atmospheric conditions, the height at which sensible heat flux is equal to zero is different. Namely, linear decrease to zero at the top of the inversion layer is assumed for Stable phase, at the NLLJ maximum for Jet phase and at the cloud base height for Stratus phase I and II. For the daytime conditions, we assume that sensible heat flux decreases

linearly with height and equals $-0.2H_0$ (where $H_0$ is the surface measured value) at the cloud top height (e.g., Stull 1988; Wood 2012). This is also verified by the analysis of turbulent fluxes measurements obtained by unmanned aerial system (UAS) ALADINA (Altstädter et al. 2015; Bärfuss et al. 2018) from 20 flights during the morning hours on 8 different days.

- Section 4.2: you have calculated the RH budget to understand whether q or T has greater impact on the RH. I agree with you regarding the premise that a moisture advection can happen but it wouldn't necessarily lead to saturation and clouds, however I disagree the way you have gone about it. The (II) and (III) terms in equation i) have the tendency terms of water vapor mixing ratio and temperature in them. Now as you have shown in the section 4.2, the tendency of temperature also depends on the advection. So I recommend you not to use the basic Clausius-Clapeyron equation, but do a classical moisture budget assuming a well-mixed boundary layer. See Caldwell et al. (2005 JAS) or Kalmus et al. (2014 J. Climate). This will enable to understand if the changes in the moisture are locally generated or a result of large-scale advection.

  In Eq. (1) we only consider what causes local changes of relative humidity. This means that we do not consider the processes causing either temperature or specific humidity changes. Also, this is not needed at this point. Each of the three terms of Eq. (1) is calculated based on available observations and there is no residual. The results obtained based on Eq. (1) indicate that the change of temperature is mainly responsible for the change in relative humidity. On the other hand, changes of specific humidity are much less pronounced and their contribution to relative humidity is substantially smaller. For this reason we concentrate on the processes responsible for the temperature change in the next step by calculating the heat budget terms (and we ignore the moisture budget as it is not important in our case). Several sentences in Section 4.2 are modified (on Pgs. 9, 10 and 11 text in red) in order to make it easier to understand our motivation to calculate relative humidity tendency, and based on that, the heat budget only.

- In the same vein, it will be great if you can show the sensible and latent heat fluxes during the study period. Thanks.

  The time series of sensible and latent heat fluxes measured during this IOP are now included in the paper in Fig. 5(d) and are disscused in the text on Pg. 8.

- It will be great if the authors can calculate the lower tropospheric stability of the study area including for the soundings launched in Accra.

  The lower tropospheric stability is defined as the difference in potential temperature $\theta$ between the 700-hPa level (free troposphere) and the surface, LTS=$\theta_{700} - \theta_0$ (Wood and Bretherton 2006, and references therein). We have calculated LTS from radiosoundings in Accra and obtained values of 12.6 K and 13.9 K for releases at 17:00 UTC and 23:00 UTC, respectively. The values of LTS for soundings at Savè are 10.2 K at 17:00 UTC and 14.9 K at 23:00 UTC. High LTS values are associated with strong, low-lying inversion for which the potential temperature increase across the capping inversion and the accumulated

static stability between this inversion and the 700-hPa reference level are large (Wood and Bretherton 2006). The potential temperature profiles up to 700 hPa ($\approx 3$ km) are shown in Fig. 12. Since our focus is on the lowest 1000 m a.g.l., because CTH is up to this height, we show potential temperature profiles in Fig. 9 in the paper only up to 1500 m a.g.l.

[Figure]

**Figure 12:** Vertical profiles of potential temperature in Accra (solid line) and Savè (dashed line) at 17:00 (red) and 23:00 UTC (green).

- The last two sections (5 and 6) are unnecessarily long and do not add any value to the manuscript. They have several repetitions and I think could be severely shortened and merged together. Thanks.

  In Section 5 discussion of our results with respect to findings from previous studies is presented, while in Section 6 our main findings are summarized. In order to keep a clear distinction between these two aspects, we keep the current structure. However, repetitions are removed and sections are shortened.

**Minor Comments:**

- Page 2, Line 8: I think you should add the in that area to this sentence. Otherwise the sentence is very generic. This doesnt apply to all of the LLC.

  Corrected.

- Page 2, Line 32: processes that instead of processes which.

  Corrected.

- Section 2.2: Please mention the frequency, temporal resolution and range resolution of the cloud radar. Same for the ceilometer and IR camera.

  The information about frequency, temporal and vertical resolution of the cloud radar, ceilometer and IR cloud camera is now included on Pg. 4, lines 27–30 and 34.

- Page 6, Line 22: Please remove the word "apparently"

  Removed.

- Page 9, line 9: "deck" not "decked".
  Corrected.

- Figure 1: Please make the distance on the x- and the y-axis the same. Currently the aspect ratio is not one.
  The aspect ratio along each axis is set to 1.

- Figure 2: The caption needs to mention what is shown in all panels. Currently it is not clear what is shown.
  The caption is changed in order to better explain what is shown in panels.

- Figure 3: Please change the color-scale of panel (b) from -50 to 10 dBz. It will be nice if you can clean up the data to only show returns from hydrometeors. The SNR can do that. It will be also nice if you can zoom-in the panel (d) and show a color-scale for the panel (d). thanks.
  The color scale for panel (b) is set as suggested. The algorithm proposed by Bauer-Pfundstein and Goersdorf (2007) is used to obtain the radar reflectivity of hydrometeors and the result is shown in Fig. 3(b). In this case, unfortunately, SNR is not helpful in cleaning up the data shown in yellow, which are the result of limitations of the proposed algorithm. The panel (d) is modified as suggested. Please note that colorbar in this case does not add much information, since IR images taken by IR cloud camera are coded in red, green and blue (RGB) components over 256 colors. That is, [R,G,B] denotes the relative contributions of red, green and blue of a given pixel, defined between 0 and 1 (with accuracy of 1/256). The color of a pixel depends on the emissivity of the corresponding sky area and consequently its brightness temperature (uncalibrated). A low cloud base, therefore, corresponds to red and a clear sky is seen as blue color. Therefore, a homogeneous low cloud deck will create a homogeneous red color image; a fragmented stratocumulus will render an image with colors ranging from red to blue (Dione et al. 2018). An additional explanation/information is provided in the caption.

- It will be nice if the authors also put this study in perspective of those done during the RADAGAST campaign (Miller and Sling, 2007 BAMS; Collow et al. 2016 QJRMS etc.)
  The RADAGAST campaign represented an international effort to measure continuously the radiative fluxes at the surface and top of the atmosphere through the seasonal progression of the West African Monsoon, which is strongly impacted by Saharan dust, biomass burning, and the development of deep convection in association with the intertropical convergence zone (Miller and Slingo 2007). The RADAGAST project was also a first international deployment of the ARM Mobile Facility (AMF), developed by Atmospheric Radiation Measurement (ARM) program by US's Department of Energy to collect data in regions of interest that can be used to improve representation of clouds and radiation in global climate models. The RADAGAST campaign took place in Niamey (Niger) in West African Sahel region during 2006. Unfortunately, we find it difficult to put our study in the perspective of studies published so far regarding the RADAGAST campaign, as there are no similarities in the studied region, topics studied (i.e. no investigation of boundary layer

processes related to low-level clouds were conducted during RADAGAST) and methods applied.

**References**

Adler, B., Babić, K., Kalthoff, K., Lohou, F., Lothon, M., Dione, C., Pedruzo-Bagazgoitia, X., and Andersen, H.: Nocturnal low-level clouds in the atmospheric boundary layer over southern West Africa: an observation-based analysis of conditions and processes, Atmos. Chem. Phys. Discuss., in review, https://doi.org/10.5194/acp-2018-775, 2018.

Altstädter, B., Platis, A., Wehner, B., Scholtz, A., Wildmann, N., Hermann, M., Käthner, R., Baars, H., Bange, J., and Lampert, A.: ALADINA  an unmanned research aircraft for observing vertical and horizontal distributions of ultrafine particles within the atmospheric boundary layer, Atmos. Meas. Tech., 8, 1627–1639, https://doi.org/10.5194/amt-8-1627-2015, 2015.

Banta, R. M., Shun, C. M., Law, D. C., Brown, W., Reinking, R. F., Hardesty, R. M., Senff, C. J., Brewer, W. A., Post, M. J., and S.Darby, L.: Observational techniques: sampling the mountain atmosphere, pp. 409–530, Chow,F. K., S. F. J. De Wekker, and B. J. Snyder, Eds., Springer, Dordrecht, https://doi.org/10.1017/CBO9781316117422, 2013.

Bärfuss, K., Pätzold, F., Altstädter, B., Kathe, E., Nowak, S., Bretschneider, L., Bestmann, U., and Lampert, A.: New setup of the UAS ALADINA for measuring boundary layer properties, atmospheric particles and solar radiation, Atmosphere, 9, https://doi.org/10.3390/atmos9010028, 2018.

Bauer-Pfundstein, M. R. and Goersdorf, U.: Target separation and classification using cloud radar Doppler-spectra, in: Proceedings of the 33rd Intern. Conf. on Radar Meteorology, vol. 11.B2, pp. 1–8, Cairns, Australia, URL https://ams.confex.com/ams/33Radar/techprogram/paper_123456.htm, 2007.

Cermak, J. and Bendix, J.: A novel approach to fog/low stratus detection using Meteosat 8 data, Atmos. Res., 87, 279–292, https://doi.org/10.1016/j.atmosres.2007.11.009, 2008.

Crewell, S. and Löhnert, U.: Accuracy of Boundary Layer Temperature Profiles Retrieved With Multifrequency Multiangle Microwave Radiometry, IEEE Trans. Geosci. Remote. Sens., 45, 2195–2201, https://doi.org/10.1109/TGRS.2006.888434, 2007.

Deetz, K., Vogel, H., Knippertz, P., Adler, B., Taylor, J., Coe, H., Bower, K., Haslett, S., Flynn, M., Dorsey, J., Crawford, I., Kottmeier, C., and Vogel, B.: Numerical simulations of aerosol radiative effects and their impact on clouds and atmospheric dynamics over southern West Africa, Atmos. Chem. Phys., 18, 9767–9788, https://doi.org/10.5194/acp-18-9767-2018, 2018.

Dione, C., Lohou, F., Lothon, M., Adler, B., Babić, K., Kathoff, N., Pedruzo-Bagazgoitia, X., Bezombes, Y., and Gabella, O.: Low Level Cloud and Dynamical Features within the Southern West African Monsoon, Atmos. Chem. Phys., submitted, 2018.

Hannak, L., Knippertz, P., Fink, A. H., Kniffka, A., and Pante, G.: Why do global climate models struggle to represent low-level clouds in the West African summer monsoon?, J. Climate, 30, 1665–1687, https://doi.org/10.1175/JCLI-D-16-0451.1, 2017.

Hunt, G. E.: Radiative properties of terrestrial clouds at visible and infrared thermal window wavelengths, Q. J. R. Meteorol. Soc., 99, 349–369, https://doi.org/doi.org/10.1002/qj.49709942013, 1973.

Knippertz, P., Fink, A. H. Schuster, R., Trentmann, J., and Schrage, J.: Ultra-low clouds over the southern West African monsoon region, Geophys. Res. Lett., 38, https://doi.org/10.1029/2011GL049278, 2011.

Löhnert, U., Turner, D., and Crewell, S.: Ground-based temperature and humidity profiling using spectral infrared and microwave observations. Part I: Simulated retrieval performance in clear-sky conditions, J. Appl. Meteorol. Clim., 48, 1017–1032, https://doi.org/10.1175/2008JAMC2060.1, 2009.

Miller, M. A. and Slingo, A.: The ARM Mobile Facility and its First International Deployment: Measuring Radiative Flux Divergence in West Africa, B. Am. Meteorol. Soc., 88, 1229–1244, https://doi.org/10.117/BAMS-88-8-1229, 2007.

Pospichal, B. and Crewell, S.: Boundary layer observations in West Africa using a novel microwave radiometer, Meteorol. Z., 16, 513–523, https://doi.org/0.1127/0941-2948/2007/0228, 2007.

Ricchiazzi, P., Yang, S., Gautier, C., and Sowle, D.: SBDART: A research and teaching software tool for plane-parallel radiative transfer in the Earth's atmosphere, B. Am. Meteorol. Soc., 79, 2101–2114, https://doi.org/10.1175/1520-0477(1998)079⟨2101:SARATS⟩2.0.CO;2, 1998.

Stull, R. B.: An Introduction to Boundary-Layer Meteorology, Kluwer Academic Publishers: Dordrecht, 1988.

Taylor, J. W., Haslett, S. L., Bower, K., Flynn, M., Crawford, I., Dorsey, J., Choularton, T., Connolly, P. J., Voigt, C., Hahn, V., Sauer, D., Dupuy, R., Brito, J., Schwarzenboeck, A., Bourriane, T., Rosenberg, P., Flamant, C., Lee, J. D., Vaughan, A. R., Hill, P. G., Brooks, B., Catoire, V., Knippertz, P., and Coe, H.: Aerosol influences on low-level clouds in the West African monsoon, Atmos. Chem. Phys., to be submitted, 2018.

Wood, R.: Stratocumulus clouds, Mon. Weather Rev., 140, 2373–2423, https://doi.org/10.1175/MWR-D-11-00121.1, 2012.

Wood, R. and Bretherton, C. S.: On the relationship between stratiform low cloud cover and lower-tropospheric stability, J. Climate, 19, 6425–6432, https://doi.org/10.1175/JCLI3988.1, 2006.